# Chamaileon: Cross-Context Binder Design with Contextualized Modeling and Mixed Sampling

**Hengyuan Cao** [1]  **Shizhuo Cheng** [1]  **Mingxuan Liu** [1]  **Weicheng Huang** [2]
**Yunhong Lu** [1]  **Chenxi Cai** [1]  **Yan Zhang** [1]  **Min Zhang** [1 3 4]

## Abstract

The rapid evolution of generative models has unlocked new potentials in protein binder design, a pivotal task in structural biology, by facilitating end-to-end generation via joint sequence-structure modeling or hallucination. However, existing approaches are predominantly implemented under a single-target, single-state assumption, limiting their ability to model multi-target or multi-state interactions required for advanced function-oriented protein design. Here, we introduce `Chamaileon`, which unifies multi-target and multi-state binder design by formulating the problem as cross-context binding landscape modeling. The framework is underpinned by a training paradigm termed *In-Context Complex Co-Design (I3CD)* for context-aware sequence-structure co-modeling. During inference, we employ *Mixture-of-Paths Sampling (MoPS)*, a scalable strategy that optimizes a single sequence across contexts while alleviating the scarcity of high-quality multi-conformational paired data. Extensive evaluation on our newly constructed benchmark, *CROSS*, demonstrates that `Chamaileon` effectively generates sequences adaptable to diverse conformational landscapes and multi-target requirements.

## 1. Introduction

*De novo* protein binder design seeks to generate protein sequences that bind specific interfaces on target proteins given structural information alone(Bennett et al., 2023; Chu et al., 2024; Winnifrith et al., 2024; Cao et al., 2022). Re-

[1]Zhejiang University, Hangzhou, China [2]Fudan University, Shanghai, China [3]Shanghai Institute for Advanced Study Zhejiang University, Shanghai, China [4]Shanghai Institute for Mathematics and Interdisciplinary Sciences, Shanghai, China. Correspondence to: Min Zhang <min_zhang@zju.edu.cn>.

*Proceedings of the 43rd International Conference on Machine Learning*, Seoul, South Korea. PMLR 306, 2026. Copyright 2026 by the author(s).

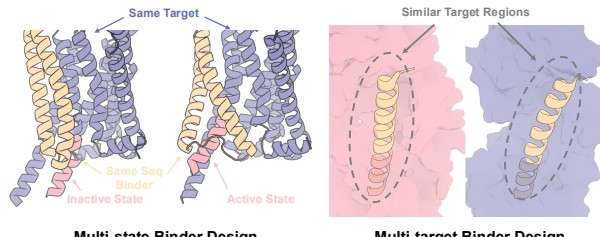

*Figure 1.* **Illustration of Cross-Context Binder Design. Multi-state (MS)** design (left) focuses on maintaining interface compatibility across the target's conformational landscape, preventing the loss of affinity when the target reshapes during functional switching. **Multi-target (MT)** design (right) focuses on multi-specificity, where a single sequence is optimized to engage similar epitopes on distinct proteins with high binding affinity.

cent advances in structure prediction(Jumper et al., 2021; Abramson et al., 2024; Passaro et al., 2025) and co-design protocols(Cho et al.; Zambaldi et al., 2024; Krishna et al., 2024; Chen et al., 2025; Pacesa et al., 2025) have improved both the reliability of in silico evaluation and the efficiency of proposing candidate binders, aided further by inference-time scaling(Anonymous, 2025). These developments make target-conditioned generation for protein-protein interactions increasingly practical.

Yet, most existing methods still treat binder design as a one-to-one problem: optimize a binder against a single target structure and a single binding objective. This assumption simplifies benchmarking, but it mismatches common function-oriented goals in biology and therapy:

First, many proteins function by switching among multiple conformational states(Kortemme, 2024; Winnifrith et al., 2024). A binder optimized against a single snapshot may not remain compatible with the same epitope as it is reshaped across functional states. In these settings, an effective binder should simultaneously accommodate state-dependent interface geometries, including cases involving large, global rearrangements during switching, which we define as *multi-state (MS) binder design*. While current approaches can sometimes tolerate modest local flexibility, explicitly designing binders to be jointly compatible with discrete, substantially different functional states remains limited and represents a

significant challenge for computational design.

Second, therapeutic efficacy improvement sometimes depends on multi-target engagement, where clinical benefit arises from coordinated modulation of distinct pathway components(Ravussin et al., 2025; Kimball et al., 2024; Chen et al., 2024; Budde et al., 2022). A single binder that satisfies multiple binding requirements, or that follows a specified selectivity pattern such as "bind protein A and protein B but not C and the others" would be valuable, which we define as *multi-target (MT) binder design*. However, most design pipelines optimize one target at a time; naive joint optimization risks nonspecificity, while sequential approaches frequently overfit one condition and fail others.

We observe that multi-state and multi-target design share the same computational structure: one sequence must satisfy multiple binding constraints with explicit trade-offs. This motivates a unified view we term **Cross-Context Binder Design**, as shown in Figure 1. Here, we define contexts as *the protein interfaces to specificically bind*, regardless of whether they come from different conformational states of the same protein *(Multi-State, MS)* or from distinct targets *(Multi-Target, MT)*. An ideal cross-context binder should be able to bind the designated interfaces while avoiding interactions with others, whether utilizing a shared surface or distinct surfaces. Under this formulation, binder design becomes a many-to-one mapping problem where success is defined by comprehensive satisfaction across these contexts rather than peak performance on any single interface.

This framing reveals a fundamental gap in current architectures. While state-of-the-art generative models excel at single-structure objectives, they lack: (i) **decoupled noise schedules for sequence and structure** to maintain sequence consistency across conformational states during joint target-binder modeling; (ii) **inference-time optimization** to navigate the trade-offs between conflicting structural constraints; and (iii) **comprehensive evaluation** that measures success across a contextual ensemble rather than on isolated snapshots. Consequently, the prevailing one-to-one paradigm cannot support the design of sophisticated molecular switches or multi-specific binders that must function across diverse conformational or target landscapes.

In this work, we present Chamaileon, a unified framework for cross-context protein binder design. Our key contributions are summarized as follows:

- We address the lack of joint representation by introducing **In-Context Complex Co-Design (I3CD)**, a paradigm that reformulates binder generation as an in-context denoising process that decouples the noise schedules for the binder sequence and structure.

- To tackle the optimization challenges inherent in multi-

objective constraints, we propose **Mixture-of-Paths Sampling (MoPS)**. MoPS enables the model to iteratively optimize a single sequence across divergent structural trajectories during inference, effectively balancing the energetic requirements of different contexts.

- Finally, we evaluate our approach on **CROSS**, a newly curated benchmark specifically designed for multi-state and multi-target binder design.

## 2. Related Work

### 2.1. Multi-state Design

Generative multi-state design methods have utilized inference-time trajectory mixing and explicit ensemble training to support multi-state compatibility. Lisanza et al. (2024) developed ProteinGenerator, a co-design diffusion model that designs fold-switching proteins by averaging sequence logits from parallel trajectories constrained by distinct structural priors. However, this heuristic aggregation often struggles to balance energetic trade-offs between conformations. Addressing sequence-level compatibility, Abrudan et al. (2025) introduced DynamicMPNN, an inverse folding model trained on CoDNaS (Monzon et al., 2016) to learn the joint conditional distribution of sequences given multiple backbones. Similarly, Jing et al. (2025) proposed ProDiT, a diffusion transformer employing "coupled structure diffusion" to co-generate distinct scaffolds. Nevertheless, these frameworks focus on modeling intrinsic scaffold heterogeneity rather than manipulating energy landscapes. While Cavanagh et al. (2026) developed "Conformational Biasing" using contrastive scoring to predict mutations that shift population distributions, existing methods lack the capability to generate modulatory binders for external regulation. We posit cross-context binder design as the missing link to precisely manipulate the target protein's energy landscape.

### 2.2. Multi-state Design Evaluation

Validating multi-state designs has shifted from standard metrics to assessing structural self-consistency. Abrudan et al. (2025) incorporated "AlphaFold Initial Guess" (AFIG) (Bennett et al., 2023) to measure state-specific refoldability, while Jing et al. (2025) utilized Chai-1 co-folding to verify distinct allosteric geometries. For dynamics, Cavanagh et al. (2026) demonstrated that likelihood-based "bias scores" correlate with conformational occupancy, and Guo et al. (2024) employed MD simulations to confirm energy minima populations. However, these frameworks evaluate scaffolds in isolation. More recently, (Zhu) has extended ensemble prediction to complex, enabling multi-state binder design, but is not validated in multi-target binder design tasks. We posit that evaluation must extend to cross-context binder design, ensuring joint compatibility with target contexts.

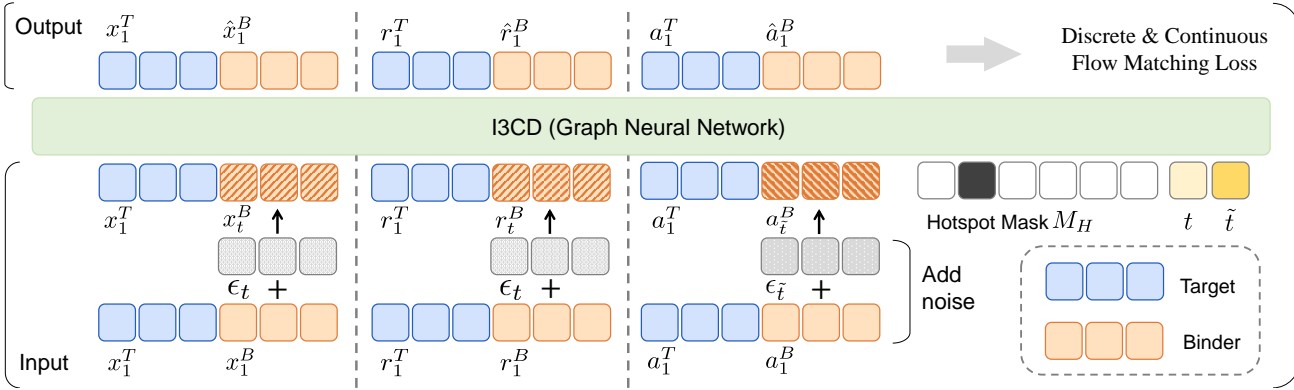

*Figure 2.* **Training pipeline of I3CD.** We concatenate clean target signals (blue tokens) with noisy binder signals (orange tokens) and feed them into the proposed I3CD paradigm to learn the joint dependencies of target and binder via discrete and continuous flow matching. Crucially, **the binder's sequence and structure are corrupted using decoupled noise schedules** $(t, \tilde{t})$, enabling the model to effectively capture the intricate sequence-structure interplay and facilitating future cross-context binder design.

## 2.3. Inference-time Guidance

Current approaches start to integrate search and optimization directly into sampling, unifying priors with test-time compute. Anonymous (2025) proposed a flow-matching framework employing MCTS and Feynman-Kac steering (Singhal et al., 2025) to optimize rewards like predction confidence. Similarly, Lisanza et al. (2024) incorporated auxiliary potentials to guide sequence diffusion. However, existing protocols remain restricted to single rigid targets or structural objectives. We extend these strategies to a multi-objective framework, addressing the complex trade-offs required for multi-state compatibility.

## 3. Preliminary

**Discrete Flow Models (DFMs).** DFMs (Campbell et al., 2024) extend the flow matching framework to discrete state spaces $x \in \{1, \dots, S\}$ by leveraging Continuous Time Markov Chains (CTMCs). Instead of the vector field used in continuous flow models (Lipman et al., 2023), the dynamics of the probability flow $p_t(x)$ are governed by a time-dependent rate matrix $R_t \in \mathbb{R}^{S \times S}$ via the Kolmogorov forward equation. To make training tractable, DFMs adopt the conditional flow matching paradigm, where a conditional probability path $p_{t|1}(x_t|x_1)$ linearly interpolates between a data sample $x_1$ and a noise distribution. Specifically, for protein sequences, a masking interpolant is commonly used:

$$p_{t|1}(x_t \mid x_1) = \text{Cat}\left(t\delta_{x_t, x_1} + (1-t)\delta_{x_t, M}\right), \quad (1)$$

where Cat denotes the Categorical distribution, $M$ is an absorbing mask state, and $\delta_{i,j}$ is the Kronecker delta. The core insight of DFMs is that the intractable marginal rate matrix can be parameterized as the expectation of a conditional rate matrix $R_t(x_t, j|x_1)$ over the posterior:

$$R_t^\theta(x_t, j) = \mathbb{E}_{p_{1|t}^\theta(x_1|x_t)}\left[R_t(x_t, j|x_1)\right], \quad (2)$$

where the conditional rate matrix is derived as $R_t(x_t, j|x_1) = \frac{\delta_{j, x_1}}{1-t}\delta_{x_t, M}$. The model is parameterized by a denoising network $p_{1|t}^\theta(x_1|x_t)$ trained via the standard categorical cross-entropy loss:

$$\mathcal{L}_{\text{ce}} = \mathbb{E}_{t \sim \mathcal{U}(0,1), x_1 \sim p_{\text{data}}, x_t \sim p_{t|1}}\left[-\log p_{1|t}^\theta(x_1|x_t)\right].$$
$$(3)$$

During inference, a discrete sequence trajectory is iteratively generated by simulating the CTMC using Euler integration steps with the learned rate matrix:

$$x_{t+\Delta t} \sim \text{Cat}\left(\delta_{x_t, x_{t+\Delta t}} + R_t^\theta(x_t, x_{t+\Delta t})\Delta t\right). \quad (4)$$

For more details, please refer to Appendix A.

**Multimodal Flows for Protein Co-Design.** A protein structure-sequence pair $\mathbf{P}$ with $D$ residues can be represented as $\{P^d = (x^d, r^d, a^d)\}_{d=1}^D$, where $x^d \in \mathbb{R}^3$ denotes the C$\alpha$ translation, $r^d \in \text{SO}(3)$ is the rotation matrix, and $a^d \in \{1, \dots, 20, M\}$ represents the amino acid type (including a mask token $M$). Given that these variables reside on different manifolds (Euclidean, Riemannian, and discrete), multimodal flow matching is adopted as the generative framework. To facilitate the forward/inverse-folding process, the dynamics of structure and sequence are decoupled via independent time schedules, $t$ and $\tilde{t}$. Under this formulation, the network takes the noisy state $\mathbf{P}_{t, \tilde{t}}$ as input and predicts the denoised translations $\hat{x}_1(\mathbf{P}_{t, \tilde{t}}; \theta)$, rotations $\hat{r}_1(\mathbf{P}_{t, \tilde{t}}; \theta)$, and amino acid distribution $p_\theta(a_1 \mid \mathbf{P}_{t, \tilde{t}})$. Training is performed by minimizing the combined flow matching loss, which corresponds to a denoising objective:

$$\mathbb{E}\left[\sum_{d=1}^D \frac{\left\|\hat{x}_1^d(\mathbf{P}_{t, \tilde{t}}; \theta) - x_1^d\right\|^2}{1-t} - \log p_\theta\left(a_1^d \mid \mathbf{P}_{t, \tilde{t}}\right)\right.$$
$$\left. + \frac{\left\|\log_{r_t^d}\left(\hat{r}_1^d(\mathbf{P}_{t, \tilde{t}}; \theta)\right) - \log_{r_t^d}\left(r_1^d\right)\right\|^2}{1-t}\right], \quad (5)$$

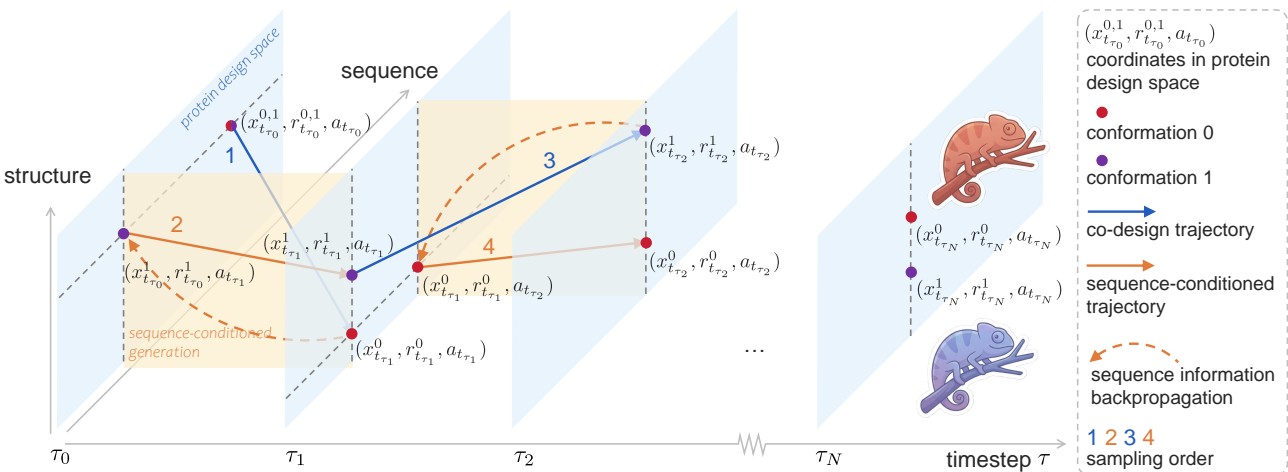

*Figure 3.* **Illustration of MoPS.** By leveraging forward folding as a bridging mechanism, MoPS interleaves the sampling trajectories of different conformations. This enables simultaneous sequence-structure co-design across multiple binder states. The name `Chamaileon` embodies our vision of AI-designed binders as chameleons, capable of adaptively adjusting their conformations in response to the context.

the model inputs $t$ and $\tilde{t}$ are omitted for simplicity.

# 4. Chamaileon

To address the critical yet underexplored challenge of cross-context binding, we present a unified framework encompassing training, sampling, and evaluation. Inspired by in-context generation in computer vision, we introduce a novel training paradigm in Section 4.1, which directly injects target information into the context of the binder denoising process. Furthermore, to enable multi-conformation binder sequence-structure co-design, we propose a method that unifies the denoising trajectories of multiple conformations in Section 4.2. Finally, Section 4.3 details the construction of our training dataset for in-context generation and introduces our evaluation benchmark for multi-state binder design.

## 4.1. In-Context Complex Co-Design

The visual in-context generation paradigm concatenates a reference condition (clean signal) with a generation target (noisy signal) and uses the self-attention mechanism to guide the synthesis process. This formulation enables the model to capture condition-target dependencies implicitly, and has proven to be highly effective for controllable image generation (Wu et al., 2025; Cao et al., 2025).

Motivated by this unified processing philosophy, we propose **I**n-**C**ontext **C**omplex **C**o-**D**esign (**I3CD**), a novel paradigm that formulates protein design as an in-context generation problem over multimodal data. Specifically, as illustrated in Figure 2, the training process begins with the clean target signals $\mathbf{T}_{1,1} = \{x_1^{T,d'}, r_1^{T,d'}, a_1^{T,d'}\}_{d'=1}^{D'}$ and binder signals $\mathbf{B}_{1,1} = \{x_1^{B,d}, r_1^{B,d}, a_1^{B,d}\}_{d=1}^{D}$. To construct the training input, we perturb the binder's sequence

and structure using independent noise schedules to obtain $\mathbf{B}_{t,\tilde{t}} = \{x_t^{B,d}, r_t^{B,d}, a_{\tilde{t}}^{B,d}\}_{d=1}^{D}$, while keeping the target signals clean. By decoupling noise injection for sequence and structure, this strategy empowers the model to capture their intricate interplay under varying noise intensities. This capability supports flexible tasks such as forward and inverse folding, while *laying a robust foundation for future cross-context binder design*. Subsequently, the corrupted binder $\mathbf{B}_{t,\tilde{t}}$ is concatenated with the clean target $\mathbf{T}_{1,1}$ across three modalities, translation, rotation, and amino acid types. Combined with the hotspot mask $M_H$ and time embeddings $t$ and $\tilde{t}$, these signals are fed into the model to predict the denoised binder signals $\hat{\mathbf{B}}_{1,1} = \{\hat{x}_1^{B,d}, \hat{r}_1^{B,d}, \hat{a}_1^{B,d}\}_{d=1}^{D}$. Following Equation (6), the model is optimized via a hybrid discrete and continuous flow matching objective, formulated as:

$$\mathbb{E}\left[\sum_{d=1}^{D} \frac{\left\|\hat{x}_1^{B,d}\big(\mathbf{B}_{t,\tilde{t}} \mid \mathbf{T}_{1,1}, M_H; \theta\big) - x_1^d\right\|^2}{1-t}\right.$$
$$- \log p_\theta\big(a_1^d \mid \mathbf{B}_{t,\tilde{t}}, \mathbf{T}_{1,1}, M_H\big)$$
$$\left. + \frac{\left\|\log_{r_t^d}\big(\hat{r}_1^d\big(\mathbf{B}_{t,\tilde{t}} \mid \mathbf{T}_{1,1}, M_H; \theta\big)\big) - \log_{r_t^d}\big(r_1^d\big)\right\|^2}{1-t}\right].$$
$$\tag{6}$$

## 4.2. MoPS for Cross-Context Binder Design

The most intuitive approach to cross-context binder design is training an end-to-end model that maps multiple target inputs to a single binder sequence with multiple conformations. However, this paradigm faces two formidable challenges. **1. Scarcity of multi-state training data.** Data capturing structural ensembles is severely limited compared to static structures. While the PDB houses vast single chains,

only a fraction (*e.g.*, ∼11,800 NMR ensembles) represent structural diversity, covering just 21% of CATH superfamilies (Abrudan et al., 2025). This scarcity is exacerbated when requiring high-quality targets with multi-state interactions. **2. Limited flexibility in handling variable conformations.** Designing a unified architecture to condition on variable target numbers is non-trivial. Most existing models are architecturally rigid, designed for fixed inputs, and struggle when the number of required binder states varies.

To surmount these obstacles, we propose a novel sampling strategy termed **M**ixture-**o**f-**P**aths **S**ampling (**MoPS**), which leverages the I3CD framework to efficiently handle cross-context binder design. Specifically, as illustrated in Figure 3, we discretize the generative trajectory along the time dimension into $N$ intervals, defined by timesteps $t_{\tau_0} = 0 < t_{\tau_1} < \cdots < t_{\tau_N} = 1$. At the initial timestep $t_{\tau_0}$, we initialize the coordinates for two binder conformations, denoted as $(x^0_{t_{\tau_0}}, r^0_{t_{\tau_0}}, a^0_{t_{\tau_0}})$ and $(x^1_{t_{\tau_0}}, r^1_{t_{\tau_0}}, a^1_{t_{\tau_0}})$, where the superscripts distinguish the structural states while $a^0 = a^1$ represents the shared sequence. For notational brevity, the residue index $d$ is omitted. The core of MoPS lies in an iterative process that alternates between co-design and sequence-conditioned generation. In the first interval (from $t_{\tau_0}$ to $t_{\tau_1}$), we advance conformation 0 via I3CD co-design (joint sequence-structure denoising) to obtain the state at $t_{\tau_1}$ with the following update rules

$$x^m_{t+\Delta t} = x^m_t + \frac{\hat{x}^m_1(\mathbf{B}^m_{t,t} \mid \mathbf{T}^m_{1,1}, M_H; \theta) - x^m_t}{1 - t} \cdot \Delta t$$
$$r^m_{t+\Delta t} = \exp_{r^m_t}(\Delta t \cdot c \cdot \log_{r^m_t}(\hat{r}^m_1(\mathbf{B}^m_{t,t} \mid \mathbf{T}^m_{1,1}, M_H; \theta)))$$
$$a^m_{t+\Delta t} \sim \mathrm{Cat}\Bigg(\delta\{a^m_t, a^m_{t+\Delta t}\}$$
$$+ \mathbb{E}_{p_\theta(a^m_1 \mid \mathbf{B}^m_{t,t}, \mathbf{T}^m_{1,1}, M_H)}\left[R_t(a^m_t, a^m_{t+\Delta t} \mid a^m_1)\right] \cdot \Delta t\Bigg)$$
$$\mathbf{B}^m_{t,t} = (x^m_t, r^m_t, a^m_t), \tag{7}$$

where $m$ represents current state index and $c$ is a constant for improving sample quality following Campbell et al. (2024). The resulting sequence $a^0_{t_{\tau_1}}$ is then extracted and used as a condition to update conformation 1 via sequence-conditioned generation (forward folding), yielding the coordinates for conformation 1 at $t_{\tau_1}$ with the following rules,

$$x^n_{t+\Delta t} = x^n_t + \frac{\hat{x}^n_1(\mathbf{B}^n_{t,1} \mid \mathbf{T}^n_{1,1}, M_H; \theta) - x^n_t}{1 - t} \cdot \Delta t$$
$$r^n_{t+\Delta t} = \exp_{r^n_t}(\Delta t \cdot c \cdot \log_{r^n_t}(\hat{r}^n_1(\mathbf{B}^n_{t,1} \mid \mathbf{T}^n_{1,1}, M_H; \theta)))$$
$$a^n_{t+\Delta t} = \hat{a}^m_1$$
$$\mathbf{B}^n_{t,1} = (x^n_t, r^n_t, \hat{a}^m_1)$$
$$n = (m+1)\%K, \tag{8}$$

where $\hat{a}^m_1$ represents the predicted amino acid types of preceding co-design process, and $K$ denotes the number of all

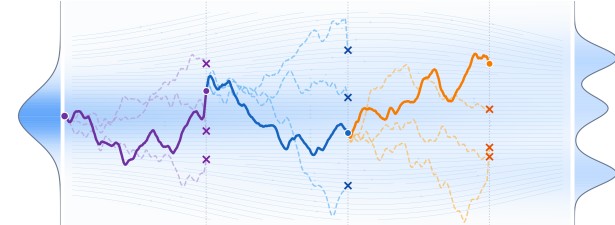

*Figure 4.* **Illustration of beam search.** In each time interval, the strategy retains only the highest-scoring candidate (solid line) while discarding less likely paths (dashed lines), ensuring efficient navigation toward the global optimum.

conformations. Consequently, at $t_{\tau_1}$, both conformations possess distinct structures but share the same updated sequence. In the subsequent interval ($t_{\tau_1}$ to $t_{\tau_2}$), we alternate the roles: conformation 1 drives the co-design process to update the sequence and its own structure, while conformation 0 is updated via forward folding based on the new sequence. This alternating process repeats until both conformations reach the final time step $t_{\tau_N}$. The primary advantage of this strategy is that it ensures *the shared sequence is iteratively optimized against the respective structural constraints of both conformations, guaranteeing that the final designed sequence is compatible with the entire conformational ensemble.* Furthermore, MoPS can be extended to the cross-context binder design task involving an arbitrary number of binder conformations, as detailed in Appendix B.

To further enhance sample quality, we incorporate beam search into the co-design phase of MoPS. As illustrated in Figure 4, this strategy involves iteratively selecting the top-performing candidates at each step to serve as the starting points for the subsequent iteration. Implementing beam search requires inherent stochasticity in the sampling outcomes. While the amino acid types naturally possess this property as they are sampled from a categorical distribution, the structural data requires additional treatment. To introduce stochasticity into the structure generation, we extend the flow matching process for the translation modality from an Ordinary Differential Equation (ODE) formulation to a Stochastic Differential Equation (SDE) framework, following Song et al. (2021); Liu et al. (2025); Geffner et al. (2025). Since our model learns a vector field mapping from a Gaussian distribution to the data distribution within the translation modality, the relationship between the predicted translation $\hat{x}_1(\mathbf{B}_{1,1} \mid \mathbf{T}_{1,1}, M_H; \theta)$ and the score $s(x_t) := \nabla_{x_t} \log p(x_t)$ is defined as follows:

$$s(x_t; \theta) = -\frac{x_t - t\hat{x}_1(\mathbf{B}_{1,1} \mid \mathbf{T}_{1,1}, M_H; \theta)}{(1-t)^2}. \tag{9}$$

Drawing theoretical grounding from the Fokker-Planck equation, we introduce a diffusion term while simultaneously correcting the drift term. This modification allows

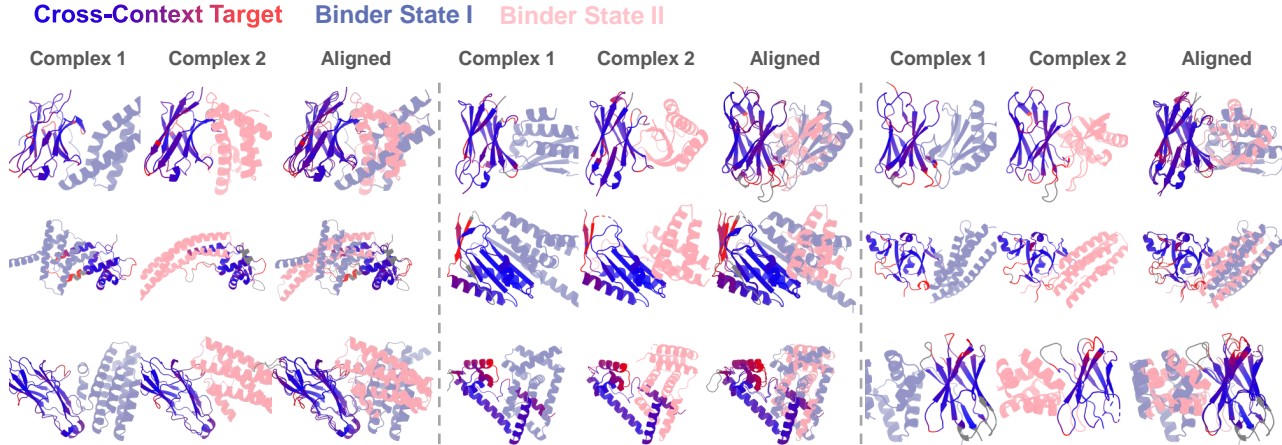

*Figure 5.* **Qualitative results of `Chamaileon` on cross-context binder design.** For each case, the target protein is shown in deep blue-red spectrum with warmer color (red) indicating higher structural deviations, while the designed binder is shown in **Binder State I** (blue) and **Binder State II** (pink). The "Aligned" column displays the superposition of both states, with structures aligned based on the target backbones to highlight how binder conformations adapt different contexts.

us to convert the originally deterministic sampling of the translation modality, as defined in Equation (7) and Equation (8), into a stochastic sampling process governed by Euler-Maruyama discretization:

$$x^m_{t+\Delta t} = x^m_t + \left(v^m_t + \frac{\sigma^2_t}{2}s^m(x^m_t; \theta)\right) \cdot \Delta t + \sigma_t\sqrt{\Delta t}\epsilon$$

$$x^n_{t+\Delta t} = x^n_t + \left(v^n_t + \frac{\sigma^2_t}{2}s^n(x^n_t; \theta)\right) \cdot \Delta t + \sigma_t\sqrt{\Delta t}\epsilon,$$

$$v_t = \frac{\hat{x}_1 - x_t}{1-t}$$

(10)

where $\epsilon \sim \mathcal{N}(0, \boldsymbol{I})$, and $\sigma_t$ represents the noise scale at different time steps. For more details on converting the ODE to the SDE, please refer to Appendix C.

Leveraging the stochastic nature introduced by the randomization described above, we can seamlessly integrate a beam search strategy into the MoPS co-design process. This integration allows for a more robust exploration of the conformational landscape by maintaining multiple potential hypotheses simultaneously. To implement this, we systematically partition the generation trajectory along the temporal dimension into $N'$ distinct intervals, formally denoted as $t_{\tau'_0} = 0 < t_{\tau'_1} < \cdots < t_{\tau'_N} = 1$. The search process operates iteratively: at the commencement of each time interval, we initialize the system with a single optimal sample point. From this anchor, we propagate $L$ parallel trajectories via random sampling to explore potential structural evolutions. Upon reaching the interval's conclusion, we rigorously evaluate the endpoints of these $L$ paths, selecting only the highest-performing candidate to serve as the seed for the subsequent phase. This step effectively acts as a

pruning mechanism, ensuring that computational resources are focused solely on the most promising structural hypotheses. To quantify performance, we analyze the predicted sequence and structural features for all $L$ candidate trajectories at every interval boundary. We compute a suite of critical metrics for each candidate, including the inter-chain predicted Aligned Error (ipAE), binder pLDDT, and the self-consistent RMSD of the binder (binder scRMSD). Finally, to determine the optimal path, we rank the candidates based on the composite score defined below.

$$\begin{aligned} \mathcal{S}(i) = &\omega_{\text{ipae}}\frac{\max_L \text{ipae} - \text{ipae}(i)}{\max_L \text{ipae} - \min_L \text{ipae}} \\ &+ \omega_{\text{plddt}}\frac{\text{plddt}(i) - \min_L \text{plddt}}{\max_L \text{plddt} - \min_L \text{plddt}} \\ &+ \omega_{\text{rmsd}}\frac{\max_L \text{rmsd} - \text{rmsd}(i)}{\max_L \text{rmsd} - \min_L \text{rmsd}}, \end{aligned}$$

(11)

where $\omega_{\text{ipae}}$, $\omega_{\text{plddt}}$, and $\omega_{\text{rmsd}}$ denote the weights of the three metrics. The candidate yielding the highest score is identified as the optimal performer. The complete MoPS algorithm with beam search is presented in Algorithm 1.

### 4.3. (Multi-State) Complex Data Collection

To support our `Chamaileon` framework, our data curation focuses on constructing a training dataset for I3CD and establishing a benchmark for cross-context binder design.

**Training Set Construction.** To train the I3CD model, we curated a dataset of chain pairs derived from the Protein Data Bank (PDB), as shown in Figure 6 (a). Initially, we iterated through all possible dimer combinations within PDB multimers. Subsequently, we filtered the raw chain pairs

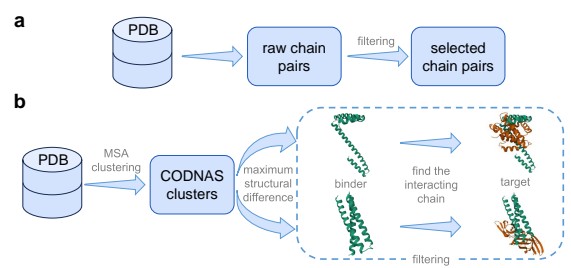

*Figure 6.* **Data collection pipeline.** (a) Construction of the I3CD training set. (b) Construction of the CROSS benchmark.

based on four distinct criteria. First, to ensure high structural quality, we excluded entries with a crystallographic resolution greater than 5 Å. Second, we imposed constraints on chain length: each chain must contain at least 16 residues, with a combined total length not exceeding 512 residues. Third, we calculated the pairwise distances between residues of the two chains, requiring at least one residue pair to have a distance of less than 8 Å, to ensure physical interaction between the two chains. Finally, we utilized AlphaFold2-Multimer to compute predicted quality metrics, specifically ipAE, binder pLDDT, and inter-chain predicted Template Modeling score (ipTM). We retained only those pairs satisfying ipAE $\leq$ 10 Å, binder pLDDT $\geq$ 80, and ipTM $\geq$ 0.5. The final filtered training set comprises 60,692 pairs.

**Cross-Context Binder Design Benchmark.** To validate Chamaileon, we introduce **CROSS** (**C**omprehensive **R**ecognition **O**f **S**pecific **S**urfaces). We leveraged CoDNaS clusters, grouped by $\geq$ 95% sequence similarity, to identify multi-state candidates, selecting the pair with maximum conformational divergence from each cluster to represent distinct states. For each structure, the interacting target chain is identified based on the highest residue contact density within the original PDB entry. Applying the same filtering criteria as the training set yields 1,867 candidates.

Structural analysis indicated a data imbalance, with 90% of target pairs exhibiting an RMSD $\leq$ 1.59 Å. To ensure benchmark diversity, we curated the final dataset by selecting the top-70 entries from the high-divergence group (RMSD > 1.59 Å) and the top-30 from the low-divergence group (RMSD $\leq$ 1.59 Å). The selection was ranked using a composite quality score, $\mathcal{S}'_{\text{item}}$, analogous to Equation (11):

$$\mathcal{S}'(i) =$$
$$\omega'_{\text{ipae}} \frac{\max_P \text{ipae} - \text{ipae}(i)}{\max_P \text{ipae} - \min_P \text{ipae}} + \omega'_{\text{plddt}} \frac{\text{plddt}(i) - \min_P \text{plddt}}{\max_P \text{plddt} - \min_P \text{plddt}}$$
$$+ \omega'_{\text{rmsd}} \frac{\max_P \text{rmsd} - \text{rmsd}(i)}{\max_P \text{rmsd} - \min_P \text{rmsd}} + \omega'_{\text{num}} \frac{\text{num}(i) - \min_P \text{num}}{\max_P \text{num} - \min_P \text{num}},$$
$$\mathcal{S}'_{\text{item}} = \min_{c \in \mathcal{C}} \mathcal{S}'(c)$$

(12)

where num denotes the count of residue pairs between the

target and binder within 8 Å, $P$ represents the set of all candidate pairs, and $\mathcal{C}$ denotes the set of conformations for a given item (taking the minimum score ensures quality across all states). Refer to Appendix E for more details.

# 5. Experiments

**Task.** We evaluate the effectiveness of our method on the task of cross-context binder design. Specifically, this task involves sequence-structure co-design where the objective is to generate a single binder sequence capable of binding to multiple distinct targets. Crucially, this single sequence must adopt different conformations (structures) corresponding to the specific binding context of each target.

**Training.** To tackle this problem, Chamaileon first leverages the I3CD paradigm for single-state binder design, then generalizes to the cross-context setting using the MoPS strategy. We train the I3CD model on the 60,692 samples detailed in Section 4.3, employing the AdamW optimizer with dynamic batching based on chain length. Experiments are performed on 8 NVIDIA A100 GPUs (80GB). The model is trained for a total of 88 epochs.

**Benchmark & Metrics.** We utilize the CROSS dataset described in Section 4.3 as our primary benchmark. Following the protocol established by Anonymous (2025), a designed binder is considered successful if it meets three distinct criteria when evaluated by AlphaFold-Multimer: an interface predicted aligned error (ipAE) $\leq$ 10, a binder pLDDT $\geq$ 70, and a binder self-consistent RMSD (scRMSD) $\leq$ 5 Å. Successful designs for each conformation are clustered using Foldseek (van Kempen et al., 2024) to determine the number of unique successes. To quantitatively assess novelty, we calculate the TM-score (Zhang & Skolnick, 2004) between the successful designs and the reference PDB structure; a lower score indicates higher novelty.

For each entry in the CROSS dataset, we generate a single sample. We report the average ipAE, binder pLDDT, binder scRMSD, and novelty across both conformations for the successful samples. Additionally, we report the count of unique successes for each conformation (which directly equals the total success count given our single-sample generation) and the number of samples where designs for both conformations are simultaneously successful.

**Results.** Qualitative results for cross-context binder design are presented in Figure 5. Chamaileon demonstrates the capability to design a single binder that binds effectively across different contexts, encompassing both MS-type and MT-type scenarios. A detailed structural analysis of these designed binders is provided in Appendix G.

The quantitative performance of our approach is comprehensively summarized in Table 1. Given the limited explo-

*Table 1.* **Quantitative results of `Chamaileon` on cross-context binder design.** We conducted comprehensive ablation studies on `Chamaileon`, including both module and parameter ablations. The best results are highlighted in **bold**.

| | | Conformation 0 | | | | | Conformation 1 | | | | | Both Success |
|---|---|---|---|---|---|---|---|---|---|---|---|---|
| | | ipAE | binder pLDDT | binder scRMSD | Unique Success | novelty | ipAE | binder pLDDT | binder scRMSD | Unique Success | Novelty | |
| **Module Ablation** | w/o MoPS | 6.26 | 82.9 | 1.83 | **17** | 0.500 | 6.27 | 82.9 | 2.49 | 2 | 0.863 | 2 |
| | w/o beam search | 5.04 | 88.2 | **1.64** | 5 | **0.363** | 6.18 | 83.9 | **1.76** | 8 | **0.304** | 5 |
| | full version | **4.32** | **90.6** | 1.96 | 8 | 0.437 | **4.23** | **88.7** | 2.07 | **10** | 0.588 | **7** |
| **Beam Search Candidate Number** | 1 | 5.27 | 88.7 | 2.39 | 7 | 0.385 | 4.78 | 89.5 | 1.77 | 6 | **0.297** | 5 |
| | 2 | **4.15** | 88.8 | **1.10** | 7 | **0.381** | 4.36 | **89.8** | 1.59 | 7 | 0.461 | 5 |
| | 4 | 4.32 | **90.6** | 1.96 | **8** | 0.437 | **4.23** | 88.7 | 2.07 | **10** | 0.588 | **7** |
| **Beam Search Frequency** | 250 | **4.23** | 89.8 | **1.45** | 9 | 0.494 | 4.97 | 87.1 | 1.78 | **11** | 0.447 | **8** |
| | 100 | 5.02 | 89.2 | 1.90 | **10** | 0.431 | 4.94 | **89.6** | **1.70** | 9 | 0.266 | **8** |
| | 50 | 4.32 | **90.6** | 1.96 | 8 | 0.437 | **4.23** | 88.7 | 2.07 | 10 | 0.588 | 7 |
| **MoPS Frequency** | 250 | - | - | - | 0 | - | 5.30 | 84.8 | 1.70 | **19** | **0.310** | 0 |
| | 100 | 5.64 | 86.4 | 1.88 | **17** | 0.473 | **2.21** | **91.7** | **1.11** | 1 | 0.928 | 1 |
| | 50 | **2.98** | **92.3** | 2.19 | 1 | 0.903 | 4.93 | 87.5 | 2.12 | 15 | 0.546 | 1 |
| | 20 | 5.14 | 88.1 | **1.81** | 13 | **0.398** | 4.10 | 90.1 | 2.07 | 9 | 0.510 | **7** |
| | 10 | 4.32 | 90.6 | 1.96 | 8 | 0.437 | 4.23 | 88.7 | 2.07 | 10 | 0.588 | **7** |

ration of machine learning approaches for this specific task, we validate the effectiveness of `Chamaileon` primarily through rigorous ablation studies. In the module ablation, the "w/o MoPS" baseline represents a sequential approach: performing single-state design on conformation 0, followed by sequence-conditioned generation on conformation 1. We observe a significant bias in unique success numbders across different conformations for this baseline, indicating a fundamental failure to simultaneously account for the structural constraints imposed by both conformations. In contrast, `Chamaileon` achieves balanced unique success numbers, demonstrating that MoPS effectively synthesizes information from multiple conformations to guide the generation process. This stark contrast underscores that a holistic integration of all conformational constraints is indispensable for robust cross-context binder design. Furthermore, the improvement in "both success" numbers over the "w/o beam search" variant confirms that beam search further enhances the performance of `Chamaileon`.

We further investigate the sensitivity of `Chamaileon` to key hyperparameters, including the "beam search candidate number", the "beam search frequency" (step interval for beam search), and the "MoPS frequency" (step interval for conformation switching). The results reveal that the "beam search candidate number" is the primary driver of beam search effectiveness; increasing the candidate numbers consistently improves "both success" numbers. Conversely, performance remains relatively insensitive to variations in "beam search frequency". Regarding MoPS, decreasing the "MoPS frequency" (i.e., reducing the switching interval) progressively mitigates the discrepancy between unique success rates across conformations while increasing "both success". This suggests that minimizing the interval for conformation

switching facilitates a more effective integration and balance of information across multiple contexts.

**Comparison with Constructed Baselines.** To thoroughly demonstrate the effectiveness of `Chamaileon`, although no existing method directly addresses cross-context binder design, we constructed two baselines as described below.

- **Baseline 1: RFDiffusion + ProteinMPNN.** We generate separate binder backbones for each target using RFDiffusion, and subsequently apply ProteinMPNN to obtain per-position amino acid distributions for both. The two distributions are then combined with equal weights, from which we sample four sequences.

- **Baseline 2: BindCraft (alternating optimization).** We adapt BindCraft's four-stage optimization procedure such that gradient updates alternate between the two target contexts. For each entry, the full pipeline is executed to produce four designed binders.

Neither baseline yielded any cross-context binder that simultaneously satisfies both contexts. To ensure the fairest comparison, we report the best per-context results across all designs for each baseline, and contrast them against the mean performance of `Chamaileon`'s successful designs.

We attribute the failure of Baseline 1 to the inherently low probability of identifying a shared binder backbone across targets under a two-stage paradigm, whereby the sequence fusion step disrupts rather than benefits the individual binding interfaces. Baseline 2 achieves improved scRMSD by optimizing over a consistent backbone; however, the frequent alternation between targets destabilizes gradient backpropagation, preventing both ipAE and pLDDT from sur-

*Table 2.* **Quantitative results of `Chamaileon` compared with two constructed baselines.** The best results are highlighted in **bold**.

| Method | Conformation 0 | | | | Conformation 1 | | | | Both Success |
|---|---|---|---|---|---|---|---|---|---|
| | ipAE | binder pLDDT | binder scRMSD | Unique Success | ipAE | binder pLDDT | binder scRMSD | Unique Success | |
| Baseline 1 (best) | 4.85 | 84.2 | 15.5 | 0 | 4.73 | 85.3 | 13.5 | 0 | 0 |
| Baseline 2 (best) | 19.1 | 67.7 | **1.37** | 0 | 18.2 | 68.1 | 3.20 | 0 | 0 |
| `Chamaileon` (mean) | **4.32** | **90.6** | 1.96 | **8** | **4.23** | **88.7** | **2.07** | **10** | **7** |

passing the acceptance thresholds. The failure of these two carefully engineered baselines underscores the inherent difficulty of cross-context binder design, and `Chamaileon`'s ability to attain non-trivial success rates demonstrates its effectiveness in addressing this challenge.

## 6. Conclusion

In this work, we introduced **Chamaileon**, a unified framework for cross-context protein binder design that transcends the traditional single-target, single-state paradigm. By decoupling the noise schedules for the binder sequence and structure, our proposed **I3CD** paradigm is able to effectively maintain sequence consistency across multiple conformations. To overcome the scarcity of multi-conformational data, we developed **MoPS**, a scalable inference-time sampling strategy that iteratively optimizes a single sequence across multiple structural contexts. Furthermore, we established **CROSS**, a rigorous benchmark specifically designed to evaluate binder adaptability across diverse conformational and target landscapes. Our experimental results demonstrate that `Chamaileon` can successfully generate high-quality binders that satisfy multi-objective constraints, providing a practical solution for designing proteins with complex functional requirements.

Looking ahead, this framework can be extended from discrete states to continuous conformational landscapes and integrate more refined biophysical priors to further enhance interface complementarity. Notably, **cross-context modeling represents a pivotal step toward the "programmable" design of advanced modulatory effects and multi-specific therapeutics**, providing a robust computational foundation for the next generation of function-oriented protein design, paving the path for allosteric modulation, broad-spectrum neutralization, and conformational ensemble manipulation.

## Acknowledgments

This work was supported by the National Major Science and Technology Projects (the grant number 2022ZD0117000) and the National Natural Science Foundation of China (grant number 62202426). We thank Shanghai Institute for Mathematics and Interdisciplinary Sciences (SIMIS) for their financial support. This research was funded by SIMIS under grant number [SIMIS-ID-2025-AD]. The authors are grateful for the resources and facilities provided by SIMIS, which were essential for the completion of this work.

## Impact Statement

This paper presents work whose goal is to advance the field of Machine Learning. There are many potential societal consequences of our work, none which we feel must be specifically highlighted here.

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

# A. Discrete Flow Models

This section presents a comprehensive derivation of the Discrete Flow Models (DFMs) framework. As proposed by Campbell et al. (2024), the concepts of continuous flow matching are extended to discrete state spaces by mapping vector fields to rate matrices in Continuous Time Markov Chains (CTMCs).

## A.1. Dynamics on Discrete State Spaces

In continuous flow matching, the evolution of a probability density path $p_t(x)$ is described by a continuity equation (Liouville equation) involving a time-dependent vector field. For discrete data $x \in \{1, \ldots, S\}$, the analog to the continuous space Fokker-Planck (or continuity) equation is the **Kolmogorov Forward Equation** (also known as the Master Equation).

Let $p_t \in \mathbb{R}^S$ denote the probability mass vector at time $t$, where the $k$-th entry represents $p_t(x = k)$. The dynamics of $p_t$ are governed by a time-dependent rate matrix $R_t \in \mathbb{R}^{S \times S}$, where $R_t(i, j)$ represents the instantaneous rate of transitioning from state $i$ to state $j$ (for $i \neq j$). The time evolution is given by:

$$\frac{d}{dt}p_t(x) = \sum_{j \neq x} \left( p_t(j)R_t(j, x) - p_t(x)R_t(x, j) \right). \tag{13}$$

This equation describes the conservation of probability mass: the change in probability at state $x$ equals the total inflow from all other states $j$ minus the total outflow from $x$. The objective of discrete flow matching is to learn a parametric rate matrix $R_t^\theta$ that generates a desired probability path $p_t(x)$ transforming a simple noise distribution $p_0$ (at $t = 0$) to the data distribution $p_{\text{data}}$ (at $t = 1$).

## A.2. Conditional Flow Matching

Directly modeling the marginal probability path $p_t(x)$ is intractable. Following the flow matching paradigm, the marginal path is constructed as a mixture of simple *conditional* paths defined per data sample $x_1$. A conditional probability path $p_{t|1}(x_t|x_1)$ is defined to interpolate between a noise distribution at $t = 0$ and a specific data point $x_1$ at $t = 1$.

The **Masking Interpolant** is utilized in this framework. Let $M$ be a special absorbing mask token. The conditional probability path is defined as:

$$p_{t|1}(x_t \mid x_1) = t\delta_{x_t, x_1} + (1 - t)\delta_{x_t, M}. \tag{14}$$

Intuitively, this implies that at time $t$, a token is revealed as the true data $x_1$ with probability $t$, and remains masked with probability $1 - t$.

To simulate this process, the corresponding *conditional rate matrix* $R_t(x_t, j|x_1)$ that generates $p_{t|1}(x_t|x_1)$ is required. By substituting the masking interpolant into the Kolmogorov equation, the analytical form of this rate matrix is derived. Specifically, probability mass must flow from the mask state $M$ to the data state $x_1$. The rate of this transition is given by:

$$R_t(x_t, j|x_1) = \frac{\delta_{j, x_1}}{1 - t}\delta_{x_t, M}. \tag{15}$$

This indicates that if the current state $x_t$ is the mask $M$, it transitions to the target $x_1$ with a rate of $\frac{1}{1-t}$. If $x_t$ is already unmasked (i.e., $x_t = x_1$), the rate is zero, and the state remains absorbing.

## A.3. Marginal Rate Parameterization and Training

While the conditional rate $R_t(\cdot|x_1)$ depends on the unknown target $x_1$, the *marginal* rate matrix $R_t(x_t, j)$ (which drives the unconditional flow $p_t$) can be expressed as the expectation of the conditional rate over the posterior distribution of the clean data:

$$R_t(x_t, j) = \mathbb{E}_{x_1 \sim p_{1|t}(x_1|x_t)} \left[ R_t(x_t, j|x_1) \right]. \tag{16}$$

This intractable true posterior $p_{1|t}(x_1|x_t)$ is approximated using a neural network $p_{1|t}^\theta(x_1|x_t)$, which predicts the clean data $x_1$ given a noisy input $x_t$ and time $t$. Consequently, the parameterized marginal rate matrix becomes:

$$R_t^\theta(x_t, j) = \mathbb{E}_{x_1 \sim p_{1|t}^\theta(x_1|x_t)} \left[ \frac{\delta_{j, x_1}}{1 - t}\delta_{x_t, M} \right] = \frac{p_{1|t}^\theta(j|x_t)}{1 - t}\delta_{x_t, M}. \tag{17}$$

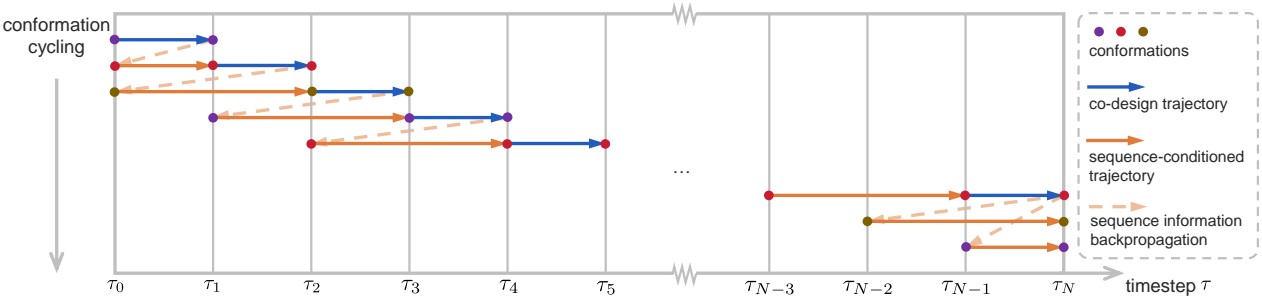

*Figure 7.* **Scalability of MoPS with respect to the number of conformations.** By cyclically alternating among target states, MoPS extends binder co-design to an arbitrary number of conformations without incurring additional computational overhead.

This formulation reveals that learning the rate matrix is equivalent to learning a denoising model. Therefore, the network $p_{1|t}^\theta$ is trained to minimize the cross-entropy loss between the predicted distribution and the ground truth data $x_1$:

$$\mathcal{L}_{\text{ce}} = \mathbb{E}_{t\sim\mathcal{U}(0,1),x_1\sim p_{\text{data}},x_t\sim p_{t|1}(\cdot|x_1)} \left[ -\log p_{1|t}^\theta(x_1|x_t) \right]. \tag{18}$$

This objective avoids the need for complex simulations or matching exact rate values during training, significantly simplifying the optimization process.

### A.4. Sampling via Euler Integration

Once the model $p_{1|t}^\theta$ is trained, samples can be generated by simulating the CTMC starting from the noise state $x_0 = M$ at $t = 0$ and evolving to $t = 1$. The Euler method is employed to discretize the continuous time dynamics.

For a small time step $\Delta t$, the transition probability from state $x_t$ to $x_{t+\Delta t}$ is approximated by:

$$P(x_{t+\Delta t}|x_t) = \text{Cat}\left( \delta_{x_t,x_{t+\Delta t}} + R_t^\theta(x_t, x_{t+\Delta t})\Delta t \right). \tag{19}$$

Substituting the derived form of $R_t^\theta$, the sampling update rule proceeds as follows:

- If $x_t \neq M$: The state remains unchanged ($x_{t+\Delta t} = x_t$) since the rate is zero.
- If $x_t = M$: The state transitions to a new category $j$ with probability $\frac{\Delta t}{1-t}p_{1|t}^\theta(j|x_t)$, or remains $M$ with probability $1 - \frac{\Delta t}{1-t}$.

This process is repeated iteratively from $t = 0$ to $t = 1$, gradually "unmasking" the sequence based on the model's predictions.

## B. Scalability of MoPS

MoPS demonstrates superior flexibility, capable of generalizing to design tasks involving an arbitrary number of conformational states. This scalability is achieved through a cyclic rolling mechanism where multiple conformations are sequentially integrated into the co-design process, as shown in Figure 7. Specifically, each conformation resumes from the endpoint of its previous co-design step. It first undergoes sequence-conditioned generation to align with the timestep of the preceding conformation, and subsequently advances one time interval via co-design. All conformations are initialized at $t = 0$. The process terminates when the first conformation reaches $t = 1$ via co-design, at which point the remaining conformations complete their trajectories to $t = 1$ through sequence-conditioned generation based on the final sequence. Crucially, for a task involving $K$ conformations, MoPS maintains computational efficiency by sampling exactly $K$ complete trajectories, incurring no additional overhead.

## C. From ODE to SDE

In this section, the theoretical derivation for extending the deterministic Ordinary Differential Equation (ODE) formulation of flow matching to a Stochastic Differential Equation (SDE) framework is presented. This extension introduces the

necessary stochasticity for the beam search strategy employed during the co-design phase. The derivation primarily follows the theoretical foundations established in Song et al. (2021), with specific adaptations for the flow matching context as discussed in Liu et al. (2025) and Geffner et al. (2025).

## C.1. Flow Matching and the Velocity Field

The flow matching framework defines a probability path $p_t(x)$ that interpolates between a source distribution $p_0(x) = \mathcal{N}(x; 0, \boldsymbol{I})$ (noise) at $t = 0$ and a target data distribution $p_1(x) = p_{\text{data}}(x)$ at $t = 1$. The interpolation path is typically defined as a conditional probability path given a data sample $x_1 \sim p_{\text{data}}(x)$:

$$x_t = (1 - t)x_0 + tx_1, \quad \text{where } x_0 \sim \mathcal{N}(0, \boldsymbol{I}). \tag{20}$$

This defines the conditional distribution $p(x_t|x_1) = \mathcal{N}(x_t; tx_1, (1 - t)^2\boldsymbol{I})$. The dynamics of the marginal distribution $p_t(x)$ are governed by the continuity equation, which can be described by an ODE:

$$dx_t = v_t(x_t)dt, \tag{21}$$

where $v_t(x_t)$ is the velocity field. In the context of optimal transport conditional flow matching, the target velocity field is defined as $v_t(x_t|x_1) = x_1 - x_0$. Expressing $x_0$ in terms of $x_t$ and $x_1$, the vector field can be rewritten as:

$$v_t(x_t|x_1) = \frac{x_1 - x_t}{1 - t}. \tag{22}$$

During inference, the neural network predicts the data sample $\hat{x}_1(\cdot; \theta)$, and the estimated velocity field is given by:

$$v_t(x_t; \theta) = \frac{\hat{x}_1(\cdot; \theta) - x_t}{1 - t}. \tag{23}$$

## C.2. Connection to Score Function

To introduce stochasticity, we must express the score function of the marginal distribution, $\nabla_{x_t} \log p_t(x_t)$, in terms of the variables available during training.

First, consider the **conditional distribution** $p(x_t|x_1) = \mathcal{N}(x_t; tx_1, (1 - t)^2\boldsymbol{I})$ of $x_t$ given a fixed data sample $x_1$. The score of this conditional distribution is given by:

$$\nabla_{x_t} \log p(x_t|x_1) = -\frac{x_t - tx_1}{(1 - t)^2}. \tag{24}$$

From the interpolation equation $x_t - tx_1 = (1 - t)x_0$, we can rewrite the conditional score in terms of the noise $x_0$:

$$\nabla_{x_t} \log p(x_t|x_1) = -\frac{(1 - t)x_0}{(1 - t)^2} = -\frac{x_0}{1 - t}. \tag{25}$$

The **marginal score** is the expectation of the conditional score over the posterior of the data $p(x_1|x_t)$. Using the identity $\nabla \log p_t(x_t) = \mathbb{E}_{x_1|x_t}[\nabla \log p(x_t|x_1)]$, we derive:

$$\nabla \log p_t(x_t) = -\frac{1}{1 - t}\mathbb{E}[x_0|x_t]. \tag{26}$$

Next, we derive the **velocity field** $v_t(x)$. By definition, the optimal velocity field matches the expected time derivative of the path:

$$v_t(x) = \mathbb{E}[\dot{x}_t|x_t = x]. \tag{27}$$

Taking the time derivative of the path $x_t = (1 - t)x_0 + tx_1$, we get $\dot{x}_t = x_1 - x_0$. Thus:

$$v_t(x) = \mathbb{E}[x_1 - x_0|x_t = x] = \mathbb{E}[x_1|x_t = x] - \mathbb{E}[x_0|x_t = x]. \tag{28}$$

We can express $x_1$ in terms of $x_t$ and $x_0$ as $x_1 = \frac{x_t - (1-t)x_0}{t}$. Substituting this into the velocity equation:

$$
\begin{aligned}
v_t(x) &= \mathbb{E}\left[\frac{x_t - (1-t)x_0}{t} - x_0 \middle| x_t = x\right] \\
&= \frac{x}{t} - \left(\frac{1-t}{t} + 1\right)\mathbb{E}[x_0|x_t = x] \\
&= \frac{x}{t} - \frac{1}{t}\mathbb{E}[x_0|x_t = x].
\end{aligned}
\tag{29}
$$

Now, we substitute the score relationship from Eq. (26), where $\mathbb{E}[x_0|x_t] = -(1-t)\nabla \log p_t(x_t)$, into the velocity equation:

$$
v_t(x) = \frac{x}{t} - \frac{1}{t}\left(-(1-t)\nabla \log p_t(x)\right).
\tag{30}
$$

This yields the fundamental relationship between the velocity field and the score function for our specific interpolation path:

$$
v_t(x) = \frac{x}{t} + \frac{1-t}{t}\nabla \log p_t(x).
\tag{31}
$$

Solving for the score, we obtain:

$$
\nabla \log p_t(x) = \frac{t}{1-t}v_t(x) - \frac{x}{1-t}.
\tag{32}
$$

During inference, we approximate $v_t = \frac{\hat{x}_1 - x_t}{1-t}$ with our neural network output $\hat{x}_1$. This allows us to compute the score required for the SDE simulation directly from the flow matching model outputs.

## C.3. From ODE to SDE via Fokker-Planck Equation

Any SDE of the form $dx_t = f(x_t, t)dt + g(t)dw$ possesses an associated Probability Flow ODE (PF-ODE) given by:

$$
dx_t = \left[f(x_t, t) - \frac{1}{2}g(t)^2\nabla_{x_t} \log p_t(x_t)\right]dt,
\tag{33}
$$

which describes the same marginal probability densities $p_t(x)$ as the SDE.

In our context, the flow matching ODE $dx_t = v_t(x_t)dt$ serves as the Probability Flow ODE. To construct a stochastic process that preserves the same marginal distributions $p_t(x)$ as the deterministic flow, we seek an SDE of the form:

$$
dx_t = \tilde{f}(x_t, t)dt + \sigma_t dw,
\tag{34}
$$

where $\sigma_t$ is a time-dependent noise scale. By equating the drift term of the PF-ODE corresponding to this SDE with the velocity field of the flow matching ODE, the following relationship is established:

$$
v_t(x_t) = \tilde{f}(x_t, t) - \frac{1}{2}\sigma_t^2\nabla_{x_t} \log p_t(x_t).
\tag{35}
$$

Solving for the modified drift term $\tilde{f}(x_t, t)$:

$$
\tilde{f}(x_t, t) = v_t(x_t) + \frac{1}{2}\sigma_t^2\nabla_{x_t} \log p_t(x_t).
\tag{36}
$$

Substituting this back into the SDE formulation yields the final stochastic differential equation:

$$
dx_t = \left(v_t(x_t) + \frac{\sigma_t^2}{2}s(x_t)\right)dt + \sigma_t dw.
\tag{37}
$$

This SDE ensures that while the trajectory of individual samples becomes stochastic, the evolution of the marginal distribution remains consistent with the original ODE training objective.

---

**Algorithm 1** Mixture-of-Paths Sampling (MoPS) with Beam Search

---

**Input:** timesteps $(t_0, t_1, \ldots, t_{N_0})$; beam search time split $\tau_0, \tau_1, \ldots, \tau_{N_1}$; beam search candidate number $L$; MoPS conformation switch frequency $freq$; current conformation index $cur = 0$; next conformation index $next = 1$; the number of conformations $C$; current timestep of all conformations $t^c = 0$.

**for** $(\tau_i, \tau_j)$ **in** $[(\tau_0, \tau_1), \ldots, (\tau_{N_1-1}, \tau_{N_1}), (\tau_{N_1}, \tau_{N_1})]$ **do**

    **for** $l$ **in** $[0, 1, \ldots, L)$ **do**

        **for** $t_m$ **in** $[t_{\tau_i}, \ldots, t_{\tau_j})$ **do**

            **if** $m \% freq == 0$ and $m \neq 0$ **then**

                **for** $t_n$ **in** $[t_{\min\{0, m-L(C-1)\}}, t_m)$ **do**

                    $\mathbf{B}^{next,l}_{t_n, t_m}$ is updated by Equations (8) and (10)

                **end for**

                $t^{next} = t_n$

                $cur = next, next = (cur + 1)\% C$

            **end if**

            $\mathbf{B}^{cur,l}_{t_m, t_m}$ is updated by Equations (7) and (10)

            $t^{cur} = t_m$

        **end for**

        $\mathcal{S}(l)$ is calculated by Equation (11)

    **end for**

    $best = \arg\max_{l \in \{0,1,\ldots,L-1\}} \mathcal{S}(l)$

    **for** $c$ **in** $[0, 1, \ldots, C)$ **do**

        $\mathbf{B}^c_{t^c, t^c} = B^{c,best}_{t^c, t^c}$

    **end for**

**end for**

**for** $k$ **in** $[0, 1, \ldots, C-1]$ **do**

    $final = (cur + k)\% C$

    **for** $t$ **in** $(t^{final}, \ldots, 1]$ **do**

        $\mathbf{B}^{final}_{t,1}$ is update by Equations (8) and (10)

    **end for**

**end for**

---

### C.4. Discretization

For numerical implementation, the Euler-Maruyama discretization scheme is applied to the derived SDE. Given a time step $\Delta t$, the update rule is:

$$x_{t+\Delta t} = x_t + \left( v_t(x_t; \theta) + \frac{\sigma_t^2}{2} s(x_t; \theta) \right) \Delta t + \sigma_t \sqrt{\Delta t} \epsilon, \tag{38}$$

where $\epsilon \sim \mathcal{N}(0, \boldsymbol{I})$. This derivation validates the sampling schema presented in Equation (10) of the main text, enabling the use of beam search for structure generation by injecting controlled noise $\sigma_t$ into the sampling process.

## D. MoPS Algorithm with Beam Search

We summarize the sampling algorithm, which combines MoPS with beam search, in Algorithm 1.

## E. More Data Collection Details

In this section, we provide further details regarding our data collection process. Figure 8 (a) illustrates the distribution of the sum of lengths for the two chains in the training set. Figure 8 (b) displays the distribution of the combined lengths of the target and binder within the candidate pool of 1,867 entries collected during the benchmark construction. Furthermore, Figure 9 (a) presents the distribution of structural differences, measured by Root Mean Square Deviation (RMSD), between the two targets for each entry in the benchmark candidate pool. Finally, Figure 9 (b) shows the structural differences between the two targets for each entry in the final CROSS benchmark.

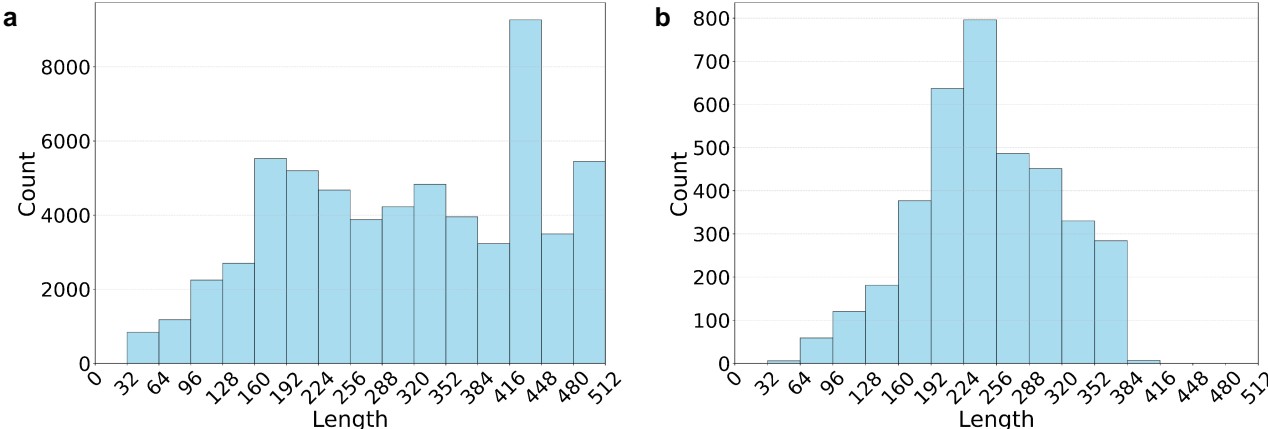

*Figure 8.* **Dimer length distributions for the training set and the benchmark candidate pool.**

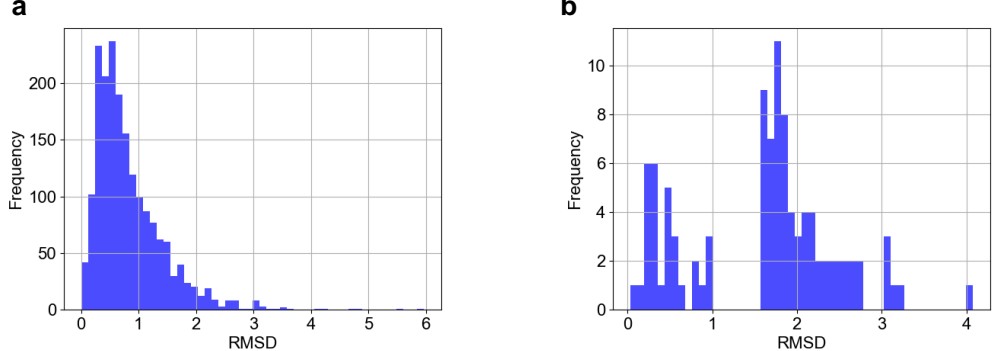

*Figure 9.* **Distributions of structural differences (RMSD) in the candidate pool and the final CROSS benchmark.**

*Table 3.* **Details of the benchmark used for the single-state binder design task.** The table lists structural information, binding specifications, and other relevant parameters.

| Target Name | PDB ID | Target Chain & Residues | Hotspot Residues | Binder Length |
|---|---|---|---|---|
| BHRF1 | 2WH6 | A2-158 | A65, A74, A77, A82, A85, A93 | 80-120 |
| H1 | 5VLI | A1-50, A76-80, A107-111, A258-322, B501-568, B580-670 | B521, B545, B552 | 40-120 |
| IL7RA | 3DI3 | B17-209 | B58, B80, B139 | 50-120 |
| IL17A | 4HSA | A17-29, A41-131, B19-127 | A94, A116, B67 | 50-140 |
| Insulin | 4ZXB | E6-155 | E64, E88, E96 | 40-120 |
| PD-L1 | 5O45 | A17-132 | A56, A115, A123 | 50-120 |
| SARS-CoV-2 RBD | 6M0J | E333-526 | E485, E489, E494, E500, E505 | 80-120 |
| TNF-$\alpha$ | 1TNF | A12-157, B12-157, C12-157 | A113, C73 | 50-120 |
| TrkA | 1WWW | X282-382 | X294, X296, X333 | 50-120 |
| VEGFA | 1BJ1 | V14-107, W14-107 | W81, W83, W91 | 50-140 |

*Table 4.* Results on the single-state binder design task. The best results are highlighted in **bold**.

| | Unique Success | Novelty |
|---|---|---|
| APM (Chen et al., 2025) | **1.2** | 0.615 |
| I3CD | 0.7 | **0.546** |

# F. Evaluating I3CD for Single-State Binder Design

In addition to the cross-context binder design task, we also validated the effectiveness of our proposed I3CD on the single-state binder design task. We selected 10 target proteins from the main results of Zambaldi et al. (2024), including BHRF1, SC2RBD, IL-7RA, PD-L1, TrkA, IL-17A, VEGF-A, insulin, H1, and TNF-$\alpha$, to serve as the benchmark for single-state binder design. Detailed specifications are provided in Table 3. Following the protocol for cross-context binder design, we define the success of a single-state binder design based on the ipAE, binder pLDDT, and binder scRMSD calculated by AlphaFold-Multimer. Specifically, we consider a sample successful if it satisfies the following criteria: ipAE $\leq$ 14, binder pLDDT $\geq$ 70, and binder scRMSD $\leq$ 5. We compared our method against APM (Chen et al., 2025), and the results are presented in Table 4.

Table 4 presents the average unique success and novelty metrics for I3CD and APM across the 10 target proteins. Consistent with the cross-context binder design task, unique success is determined by clustering successful samples using Foldseek, where the number of clusters represents the unique success count. Novelty is evaluated using the TM-Score. The results indicate that while I3CD generates samples with superior novelty compared to APM, it achieves a lower success rate. We attribute this performance gap to the smaller size of our base model, which possesses a more limited learning capacity compared to current state-of-the-art models. Specifically, our model contains 21.8 million parameters, whereas the APM model has 199.6 million, a difference of more than ninefold.

# G. Binder Structure Analysis of Cross-Context Binder Design

Analysis of the generated cross-context binders reveals a hierarchical classification of structural adaptation strategies. As illustrated in Figure 10, Chamaileon enables a single sequence to navigate the backbone plasticity required to satisfy multi-objective constraints through three distinct modes:

- **Micro Adaption**: In this mode, the binder maintains a highly conserved scaffold across different contexts. Joint compatibility is primarily achieved through subtle backbone fluctuations and localized adjustments within a nearly identical global fold, allowing the binder to tolerate minor variations in the target interface.

- **Dual-face Adaption**: The binder preserves a stable and rigid backbone fold but utilizes spatially distinct surfaces (faces) to engage with different target interfaces. This strategy enables multi-specific recognition by repurposing different regions of the same protein fold, requiring minimal structural deformation.

- **Macro-switch Adaption**: For more challenging tasks involving drastically different interface geometries, the binder undergoes large-scale backbone rearrangements or partial fold-switching. This represents the highest level of structural plasticity, where the single sequence adopts divergent conformations to optimize binding for each specific context.

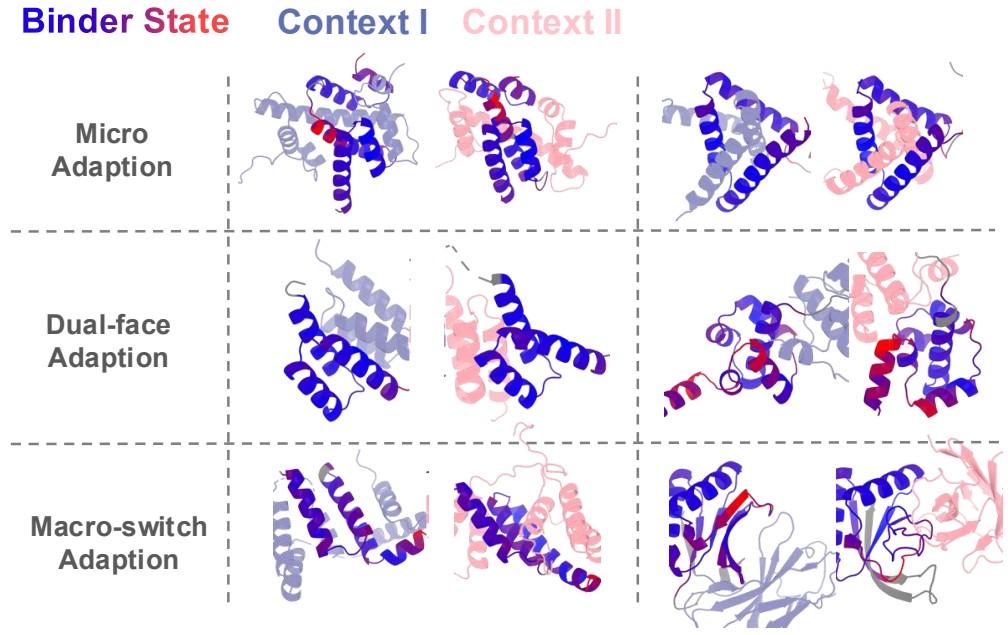

*Figure 10.* **Taxonomy of structural adaptation modes in cross-context binder design.** The figure illustrates three distinct modes of backbone plasticity: **Micro Adaption** (top), **Dual-face Adaption** (middle), and **Macro-switch Adaption** (bottom). Binder conformations for Context I (blue) and Context II (pink) are superimposed. Deep blue-red spectrum regions indicate structural deviations (RMSD) between the two binder backbones with warmer tones (red) indicating higher structural divergence.

*Table 5.* Quantitative results of `Chamaileon` on cross-context binder design across all benchmark candidates.

| | Conformation 0 | | | | | Conformation 1 | | | | | Both Success |
|---|---|---|---|---|---|---|---|---|---|---|---|
| | ipAE | binder pLDDT | binder scRMSD | Unique Success | novelty | ipAE | binder pLDDT | binder scRMSD | Unique Success | novelty | |
| `Chamaileon` | 5.71 | 83.7 | 2.19 | 83 | 0.587 | 6.07 | 83.1 | 2.20 | 100 | 0.558 | 69 |

## H. Additional Quantitative Results across All Benchmark Candidates

To further demonstrate the robustness of our framework, we conducted an additional evaluation on all 1,867 benchmark candidates. The results are presented in Table 5. The lower success rate, compared with that reported in Table 1 confirms that cross-context binder design becomes more challenging on lower-quality targets, thereby further validating the rationale behind our quality-based filtering strategy for CROSS.

## I. Active Negative Design

Our framework supports active negative design through a straightforward mechanism: we modify the score function used to rank candidates during beam search by negating the scores computed on the unwanted target, then aggregate all scores normally for ranking. Meanwhile, during MoPS, we only alternate between the two desired targets. To evaluate this capability, we selected four cases in which `Chamaileon` successfully designs binders for both contexts, and identified a third context for each case based on CoDNAS. We conducted three experiments: (i) jointly binding all three contexts during the MoPS stage; (ii) binding only two of the three contexts during the MoPS stage; and (iii) following the active-negative setting described above, in which the binder is encouraged to actively bind two contexts while explicitly avoiding binding to the remaining one. The results are reported in Table 6.

With the active/negative sampling strategy, the designed binder exhibits weaker binding to the third (unwanted) context:

*Table 6.* **Quantitative results of active negative design.** The best results are highlighted in **bold**.

| Setting | Conformation 0 | | | Conformation 1 | | | Conformation 2 | | |
|---|---|---|---|---|---|---|---|---|---|
| | ipAE | binder pLDDT | binder scRMSD | ipAE | binder pLDDT | binder scRMSD | ipAE | binder pLDDT | binder scRMSD |
| 3-context design | 4.92 | 88.7 | 2.55 | 4.67 | **90.6** | 2.01 | **4.11** | **90.9** | **1.67** |
| 2-context design | **3.54** | **92.2** | **1.09** | 4.56 | 89.2 | **1.49** | 22.2 | 74.1 | 5.31 |
| (2-act. & 1-neg.)-context design | 5.10 | 88.2 | 1.81 | **4.51** | 90.5 | 3.10 | 24.0 | 73.5 | 5.53 |

*Table 7.* **Ablation study on hyperparameters of the score function.** The best results are highlighted in **bold**.

| $\omega_{ipae}$ | $\omega_{plddt}$ | $\omega_{rmsd}$ | Conformation 0 | | | | | Conformation 1 | | | | | Both Success |
|---|---|---|---|---|---|---|---|---|---|---|---|---|---|
| | | | ipAE | binder pLDDT | binder scRMSD | Unique Success | Novelty | ipAE | binder pLDDT | binder scRMSD | Unique Success | Novelty | |
| 1.0 | 0.0 | 0.0 | 4.75 | 87.5 | 2.14 | **11** | **0.335** | 4.88 | 86.8 | 1.80 | 11 | 0.335 | **10** |
| 0.0 | 1.0 | 0.0 | 4.36 | 90.2 | 2.05 | 8 | 0.597 | 4.77 | **90.2** | 1.73 | 9 | 0.424 | 7 |
| 0.0 | 0.0 | 1.0 | 5.98 | 85.7 | **1.80** | 8 | 0.336 | 5.17 | 87.4 | **1.41** | 11 | **0.174** | 7 |
| 0.33 | 0.33 | 0.33 | 4.84 | 89.2 | 1.82 | 10 | 0.454 | 4.89 | 88.4 | 2.16 | **12** | 0.385 | **10** |
| 0.5 | 0.3 | 0.2 | **4.32** | **90.6** | 1.96 | 8 | 0.437 | **4.23** | 88.7 | 2.07 | 10 | 0.588 | 7 |

ipAE, pLDDT, and scRMSD on target 2 all degrade compared to the 2-context design baseline, confirming that this mechanism effectively repels the unwanted conformation. This demonstrates that `Chamaileon` can be readily extended to support explicit negative design without architectural modifications.

## J. Ablation Study on Hyperparameters of the Score Function

To investigate the impact of the individual weighting coefficients in the score function employed by the beam search procedure within MoPS, we conducted a series of ablation studies, with results reported in Table 7.

Shifting the weight toward a specific metric generally improves that metric while potentially degrading others. Although different weight configurations affect the final both success count, extreme settings such as $(1.0, 0.0, 0.0)$ attain a higher both success count yet exhibit noticeably weaker scRMSD and pLDDT compared with most alternative configurations. Our chosen setting strikes a balance among high quality across all three metrics (ipAE, binder pLDDT, and binder scRMSD) and a reasonable both success count, while achieving strong ipAE performance as intended by the weight design.

## K. Detailed Construction of the CROSS Benchmark

**Overview and Design Philosophy.** The CROSS benchmark is curated through a rigorous multi-stage pipeline. Starting from an initial pool of 1,867 candidates, we apply target similarity-based category balancing followed by quality-based filtering to arrive at a compact yet high-quality benchmark. This design philosophy serves two complementary purposes: it substantially reduces the computational cost of evaluation, and it ensures that each benchmark entry is of sufficient quality to yield reliable assessment. In what follows, we provide a detailed account of the construction procedure and clarify several aspects that may otherwise be subject to misinterpretation.

**Sequence Similarity Threshold.** A potential source of confusion concerns the role of the 95% sequence similarity threshold. We emphasize that this threshold is applied exclusively to the binders within each cluster derived from the CoDNAS database, rather than to the targets. Since proteins that naturally bind multiple distinct targets are exceedingly rare, we relax this constraint by treating two proteins exhibiting at least 95% sequence similarity as effectively the same binder. Once such a pair of highly similar binders is identified, their respective interacting chains are retrieved from the PDB and designated as the targets. Crucially, no sequence similarity filtering is applied to the targets themselves; consequently, the resulting target pairs may exhibit substantial divergence in both sequence and structure.

**Coverage of Multi-Target and Multi-State Scenarios.** CROSS is not intended to be an exclusively multi-state benchmark. On the contrary, during construction we deliberately increased the proportion of entries exhibiting higher target divergence to ensure broader coverage. To draw a precise distinction, multi-target refers to designing a binder that simultaneously binds

two structurally and sequentially distinct targets, whereas multi-state refers to binding different conformations of the same, or nearly identical, target. The principal distinction lies in sequence identity, which in turn manifests as varying degrees of structural divergence. Using a 95% target sequence identity threshold to separate the two categories, CROSS comprises 15 multi-target entries and 85 multi-state entries. For comparison, the initial pool of 1,867 candidates prior to filtering contains 200 multi-target and 1,667 multi-state entries, corresponding to a multi-target proportion of approximately 10.7%. The fact that this proportion is elevated to 15% in the final benchmark reflects two observations: multi-target data is naturally scarce, and we deliberately enriched CROSS to provide more meaningful coverage of the multi-target setting.

**Multi-Target Performance and Limitations.** Among the successful cases produced by `Chamaileon` on CROSS, one corresponds to a genuine multi-target scenario, in which the two targets share only 94% sequence identity yet are simultaneously bound by the designed protein with strong predicted metrics on both contexts (ipAE values of 3.32 and 3.16, pLDDT values of 93.6 and 94.1, and scRMSD values of 0.90 and 1.17, respectively). This case accounts for 1 of 7 both-success outcomes (14.3%), closely matching the multi-target proportion within CROSS itself (15%).

We acknowledge that multi-target binder design is an inherently challenging problem for which no existing end-to-end approach is available, and even straightforward adaptations of established pipelines exhibit fundamental limitations, as demonstrated by the two baselines introduced in the main paper. Our work therefore provides a feasible solution accompanied by a successful multi-target case that serves as a proof of concept. Nevertheless, we recognize considerable room for improvement, owing to the natural scarcity of evaluation data, the relatively modest size of our model, and the potentially limited capability of AlphaFold2 itself in assessing multi-target binding fidelity.

