# OpenReview forum: "Chamaileon: Cross-Context Binder Design with Contextualized Modeling and Mixed Sampling"
_ICML.cc/2026/Conference — ICML 2026 spotlight_

### Official Review · Reviewer_EyZZ · 2026-03-08

**Soundness:** 3
**Presentation:** 3
**Significance:** 3
**Originality:** 3
**Overall Recommendation:** 5
**Confidence:** 4

**Summary:**

This paper formalizes a new task of protein design with relevance to real-world applications, which requires generating protein binders specific to two different states of the target protein or two different targets. To do this, the authors propose Chamaileon, which is based on single-conditioned training, but extend to double-conditioned design via alternating the two conditions every time interval during the flow matching process. The authors further propose to integrate beam search into the generative process to enhance generation quality, with confidences from AlphaFold2-Multimer (AFM) as the evaluation proxy. For benchmarking, the authors first collect dimers from PDB to form a database of 60K complexes for training the single-conditioned model. Next, the authors cluster the proteins with 95% sequence similarity and find the pairs with most diverge conformations in the same clusters as the multiple states of the protein. In this way, the authors construct CROSS, the test set for the task. Evaluated by AFM confidences and scRMSD, the authors show that the proposed method could complete the task with observable success rates.

**Compliance With Llm Reviewing Policy:**

Affirmed.

**Final Justification:**

The rebuttal fully solved my concerns and confirmed the superiority of the proposed model. So I've raised my score.

**Key Questions For Authors:**

1. Why do we need AFM in the single-conditioned data collection? Aren't the dimers already with solved structures?
2. It looks to me that the successful designs all have similar target conformation 0 and conformation 1. How is the model performance on targets with large conformation RMSD?

**Limitations:**

yes

**Strengths And Weaknesses:**

**Strengths**

1. The paper is well written and easy to understand.
2. The authors try to tackle a new task in protein design which is more relevant to real-world applications then the conventional benchmark. The paper is complete and sound, from task construction, to model design, to data collection and evaluation.
3. The proposed method is intuitive and straightforward, which might also be useful in other fields in addition to cross-context protein design.

**Weaknesses**

1. There seems to be a mismatch between the claimed scope of cross-context protein design and the constructed testing benchmark. In task definition, cross-contexts including two different conformations of the same protein or two distinct proteins. However, in the constructed CROSS benchmark, the pairs of conditions are extracted from clusters with >95% sequence similarity, which should approximate the setting of protein binder specific to two different conformations. However, where is the setting of two distinct proteins as conditions evaluated? The authors should check the scope to avoid overclaiming.

2. If my understanding is correct, the test set contains 100 targets, but only 7 of them have successful cross-context designs, which indicate very low success rates. I'm not sure whether this is because the task is too hard or the proposed model is not so strong as a base model. Therefore I suggest at least adding one baseline from literature for comparison, as the high-level methodology of MoPS is straightforward and easily extendable to other frameworks. For example, we can exchange the target in BindCraft for every several steps to achieve this cross-context design as well.

3. Also, the beam search requires evaluation of AF2 every time, which seems to be very computational intensive. The authors should analyze the trade off between quality and running time.

---

> ### Author Rebuttal · Authors · 2026-03-31
>
> We sincerely thank the reviewer for recognizing the real-world relevance of our proposed task, the completeness of our work spanning task formulation, model design, data collection, and evaluation, as well as the potential extensibility of our method to other domains. We address each concern below.
>
> ---
>
> **[W1] Clarification on CROSS benchmark construction.**
>
> We apologize for the confusion and appreciate the opportunity to clarify. The ≥95% sequence similarity threshold mentioned in Lines 356–362 refers to the similarity among **binders** within each cluster from the CodNAS database, not the similarity among targets. Since finding a single protein that naturally binds to multiple distinct targets is extremely rare, we relax the constraint by treating two proteins with ≥95% sequence similarity as effectively the same binder. After identifying such a pair of highly similar binders, we then retrieve their respective interacting chains from the PDB as targets. Crucially, no sequence similarity filtering is applied to these targets, therefore they can be highly divergent.
>
> Regarding the specific proportions of multi-target versus multi-state cases in CROSS, please refer to our response to Reviewer 1WGH [W1], where we show that the two categories are approximately balanced. We will revise the manuscript to make this construction procedure clearer and avoid potential misunderstanding.
>
> ---
>
> **[W2] Low success rate and baseline comparisons.**
>
> We fully agree that baselines strengthen the empirical contribution. Since no existing method directly addresses cross-context binder design, we have constructed two baselines following the reviewer's suggestion:
>
> **Baseline 1: RFDiffusion + ProteinMPNN.** We generate separate binder backbones for each target via RFDiffusion, then apply ProteinMPNN to obtain per-position amino acid distributions for both. We mix the two distributions with equal weights and sample 4 sequences from the mixture.
>
> **Baseline 2: BindCraft (alternating optimization).** We modify BindCraft's four-stage optimization procedure so that gradient steps alternate between the two target contexts. For each entry, we run the full pipeline to produce 4 designed binders.
>
> Neither baseline produced any cross-context binder satisfying both contexts simultaneously. For the fairest comparison, we report **the best per-context results** across all designs for each baseline against **the mean of Chamaileon's successful designs**:
>
> | Method | ipAE_0 | pLDDT_0 | scRMSD_0 | success_0 | ipAE_1 | pLDDT_1 | scRMSD_1 | success_1 | both_success |
> |---|:---:|:---:|:---:|:---:|:---:|:---:|:---:|:---:|:---:|
> | Baseline 1 (best) | 4.85 | 84.2 | 15.5 | 0 | 4.73 | 85.3 | 13.5 | 0 | 0 |
> | Baseline 2 (best) | 19.1 | 67.7 | **1.37** | 0 | 18.2 | 68.1 | 3.20 | 0 | 0 |
> | Chamaileon (mean) | **4.32** | **90.6** | 1.96 | **8** | **4.23** | **88.7** | **2.07** | **10** | **7** |
>
> We attribute the failure of Baseline 1 to the inherently low probability of identifying a shared binder backbone across targets under a two-stage paradigm, causing the sequence fusion step to disrupt rather than benefit individual binding interfaces. Baseline 2 achieves better scRMSD by optimizing over a consistent backbone, yet frequently alternating targets destabilizes gradient back-propagation, preventing ipAE and pLDDT from surpassing acceptance thresholds. The failure of both well-engineered baselines underscores the inherent difficulty of cross-context binder design, and Chameleon's ability to achieve non-trivial success rates demonstrates its effectiveness in addressing this challenge. We will incorporate these baselines into the revised manuscript.
>
> ---
>
> **[W3] Trade-off between quality and running time.**
>
> We have added experiments analyzing this trade-off. Please refer to our response to Reviewer 1WGH [W3 & Q3], where we show that model performance steadily improves as beam search computation increases. Notably, our method takes only approximately **3.97 minutes** per design, compared to the up to **4 hours** on a single NVIDIA A100 GPU reported by Complexa (Appendix F), representing a significant advantage in computational efficiency.
>
> ---
>
> **[Q1] Why is AFM used for filtering the single-conditioned training data?**
>
> Using AFM serves to align the training data with the predictive capacity of current folding models, as these metrics provide reliable confidence estimates for predicted PPI quality, following the same practice adopted by Complexa. Additionally, this filtering step helps exclude incomplete data.
>
> ---
>
> **[Q2] Model performance on targets with large conformational differences.**
>
> Our successful designs do include cases where the two targets are distinct proteins (please refer to our response to Reviewer 1WGH [W1] for details). We acknowledge that such multi-target cases are relatively rare in real-world data, and we will include additional multi-target experiments in the revised manuscript.

---

> > ### Author Rebuttal · Reviewer_EyZZ · 2026-04-03
> >
> > Thanks for the detailed respones! They perfectly solved my concerns, and I've raised my score for acceptance!

---

> > > ### Author Response · Authors · 2026-04-05
> > >
> > > We sincerely thank the reviewer for the encouraging feedback and for the thorough evaluation throughout the review process. We will incorporate all discussed improvements into the revised manuscript.

---

### Official Review · Reviewer_dkk7 · 2026-03-11

**Soundness:** 3
**Presentation:** 2
**Significance:** 3
**Originality:** 2
**Overall Recommendation:** 4
**Confidence:** 2

**Summary:**

This paper proposes Chamaileon, a generative framework for protein binder design that models cross-context binding landscapes to support multi-target and multi-state interactions. The method introduces In-Context Complex Co-Design (I3CD) for contextualized sequence–structure modeling and Mixture-of-Paths Sampling (MoPS) to guide generation across multiple structural contexts. Experimental results on the proposed CROSS benchmark and additional evaluations suggest improvements in binding success rate and structural consistency compared to existing design approaches.

**Compliance With Llm Reviewing Policy:**

Affirmed.

**Final Justification:**

The rebuttal has addressed most of my concerns, and I will keep my positive score.

**Key Questions For Authors:**

- What is the key methodological novelty of I3CD compared to existing sequence–structure co-design frameworks?
- How well does the model generalize to targets or binding contexts not seen during training?
- Which component contributes most to the performance gains (cross-context modeling, I3CD training, or MoPS sampling)?
- How sensitive is the model to the number of structural contexts provided during training or inference?

**Limitations:**

Yes

**Strengths And Weaknesses:**

**Strengths**
- Multi-target and multi-state binder design is an important challenge in protein engineering and drug discovery.
- Modeling binding as a cross-context landscape is a natural formulation for handling multiple structural contexts.
- Results on the CROSS benchmark suggest improvements in binding success rate and structural consistency across contexts.

**Weakness**
- The paper introduces several new terms (cross-context modeling, I3CD), but the distinction from existing sequence–structure co-design or diffusion-based binder design methods is not clearly articulated.
- Most experiments are conducted on the CROSS benchmark introduced in this work, which makes it difficult to assess generalization.

---

> ### Author Rebuttal · Authors · 2026-03-31
>
> We sincerely thank the reviewer for recognizing the importance of cross-context binder design, the naturalness of our modeling formulation, and the effectiveness of our method. We address each concern below.
>
> ---
>
> **[W1 & Q1] Clarification of terminology and methodological novelty.**
>
> We appreciate the opportunity to clarify these concepts.
>
> **Cross-context modeling** (Lines 64–69) refers to the setting where a single binder sequence must satisfy multiple binding constraints with explicit trade-offs, that is, one binder can bind to multiple targets, each constituting a distinct structural "context".
>
> **I3CD (In-Context Complex Co-Design)** (Lines 201–207) draws inspiration from the in-context generation paradigm in computer vision. Given a target's context (its sequence and structure), I3CD generates the corresponding binder's sequence and structure simultaneously, hence the term "complex co-design".
>
> The key distinction between I3CD and existing diffusion-based co-design methods lies in the in-context formulation. Current approaches (e.g., Multiflow) focus on individual proteins rather than complexes, leaving open how to incorporate target information during generation. We address this by concatenating the clean target with the noisy binder and feeding them jointly into the transformer for conditional denoising. This yields a simpler architecture than diffusion-based binder design methods such as Complexa, which require separate binder/target embedding and joint pair representation extraction.
>
> ---
>
> **[W2] Evaluation beyond the CROSS benchmark.**
>
> We would like to note that the cross-context binder design task we propose is a new problem formulation for which no established benchmark previously existed, this is precisely the motivation for introducing CROSS. CROSS is a carefully curated, compact, and challenging benchmark designed to rigorously evaluate method effectiveness. For instance, as shown in our response to Reviewer EyZZ [W2], two additional baselines we evaluated on CROSS failed to produce any successful designs, while our method achieves a "both success" count of 7.
>
> Beyond CROSS, we have also conducted a larger-scale evaluation on 1,000 cross-context data pairs to further assess generalization. Please refer to our response to Reviewer JJFr [W1] for full details and results, which demonstrate that our method generalizes well to data outside the CROSS benchmark.
>
> ---
>
> **[Q2] Generalization to unseen targets or binding contexts.**
>
> This is an excellent point, and we would like to highlight an important aspect of our framework's design: our training and inference settings are intentionally mismatched. Specifically, the model is trained exclusively on single-state binder design data, while inference is performed on the cross-context binder design task. This means that, strictly speaking, none of the multi-target test cases are fully observed during training. The fact that our method still performs well in this setting demonstrates its strong generalization capability, MoPS effectively transfers a single-state binder design model to the cross-context setting, elegantly circumventing the challenge of collecting paired cross-context training data.
>
> ---
>
> **[Q3] Contribution of individual components.**
>
> We would like to clarify that the two proposed modules, I3CD and MoPS, together constitute the complete pipeline, and their relationship is complementary rather than modular in an ablatable sense. I3CD serves as the generative backbone that models single-state binder co-design, while MoPS is the inference-time strategy that extends this capability to the cross-context setting. Neither can function independently for the cross-context task: without I3CD, there is no base generative model; without MoPS, the model cannot handle multiple structural contexts simultaneously. Additionally, "cross-context modeling" refers to the overarching design philosophy guiding the entire pipeline, rather than a specific separable component.
>
> ---
>
> **[Q4] Sensitivity to the number of structural contexts.**
>
> We appreciate this question. During **training** (i.e., the I3CD training phase), only single target–binder pairs are used, no multi-context data is involved. The multi-context setting arises exclusively during **inference**, where MoPS coordinates generation across multiple targets.
>
> Regarding sensitivity to the number of contexts at inference time, please refer to our response to Reviewer JJFr [Q1], where we compare results under 2-conformation and 3-conformation settings. The results show that metrics for conformations 0 and 1 (ipAE, binder pLDDT, binder scRMSD) are slightly better in the 2-conformation setting than in the 3-conformation setting. This indicates a modest performance decrease as the number of contexts increases, which is expected: requiring a single binder to simultaneously satisfy three distinct structural contexts is inherently more challenging than satisfying two.

---

> > ### Author Rebuttal · Reviewer_dkk7 · 2026-04-03
> >
> > Thanks for your responses. The rebuttal has addressed most of my concerns, and I will keep my positive score.

---

> > > ### Author Response · Authors · 2026-04-05
> > >
> > > We thank the reviewer for the positive assessment and for acknowledging that most concerns have been addressed. We are happy to answer any follow-up questions the reviewer may have and will do our best to provide timely and thorough responses.

---

### Official Review · Reviewer_XNNu · 2026-03-16

**Soundness:** 3
**Presentation:** 3
**Significance:** 3
**Originality:** 3
**Overall Recommendation:** 4
**Confidence:** 3

**Summary:**

This paper proposes a deep learning framework for protein binder design, focusing on breaking the traditional "single-target, single-state" assumption. Borrowing concepts from computer vision, the authors introduce a unified approach for multi-target and multi-state design. They utilize a mixed sampling strategy to address the scarcity of high-quality paired data and construct a new benchmark dataset.

**Compliance With Llm Reviewing Policy:**

Affirmed.

**Final Justification:**

My concern has been addressed. I raised my score

**Key Questions For Authors:**

Regarding the evaluation metrics, I'm not entirely convinced that the current in silico metrics fully capture the model's capabilities. Would it be possible to include more standard biophysical metrics like RMSD (against real structures) or $\Delta G$?
Following up on the ablations, how can you empirically demonstrate that I3CD is more effective than traditional diffusion approaches that use joint noising?
Could you clarify how the hyperparameters for the scoring function $\mathcal{S}(i)$ were chosen during inference? What is the specific rationale behind formulating the scoring function this way?
The paper mentions that the Chamaileon model has a smaller parameter count. It would be much more comprehensive if you could also provide a comparison of training times.

**Strengths And Weaknesses:**

Strengths:
Author innovatively unify "multi-state" and "multi-target" design into a single modeling framework. Compared to single-point optimization, this naturally enhances biological realism and protein diversity.
The proposed inference strategy is quite novel. Leveraging the sequence generation trajectory to make the generated sequence satisfy multiple distinct structural constraints is a clever approach.
The introduction of the new CROSS benchmark is a solid contribution. Weighting several protein structural metrics to form a composite score seems to effectively improve the evaluation.

Weaknesses:
The evaluation somewhat lacks metrics that compare directly against real-world experimental data. While the paper uses real protein complex data during training, the inference and result evaluation phases do not use metrics that directly compare against native structures.
The ablation studies feel incomplete. There is no direct comparison between the proposed I3CD mechanism and traditional diffusion methods that add noise to sequences and structures simultaneously. Additionally, during inference, the proposed scoring function $\mathcal{S}(i)$ isn't compared against other alternative scoring methods

---

> ### Author Rebuttal · Authors · 2026-03-31
>
> We sincerely thank the reviewer for the expert and thorough review, particularly for recognizing the novelty and effectiveness of our approach, its contribution to enhancing biological realism and protein diversity, and the value of the CROSS benchmark. We address each concern in detail below.
>
> ---
>
> **[W1 & Q1] Comparison against real-world data.**
>
> We appreciate this concern. In fact, we do report a metric that compares generated designs against real-world data: the **novelty** metric in Table 1, which is based on TM-score and measures the structural similarity between successfully designed binders and entries in the PDB database (Lines 363–367).
>
> We chose TM-score over RMSD for this purpose because TM-score is a global topology metric that answers whether two proteins share the same fold, whereas RMSD is a local precision metric best suited for comparing highly similar structures at the atomic level. In our setting, designed binders and their closest PDB matches may differ in length or local details while sharing the same overall fold, a scenario where RMSD can be misleadingly large but TM-score remains informative.
>
> Regarding Rosetta $\Delta G$, its absolute values lack a rigorous quantitative correspondence to true thermodynamic binding free energies, and cross-target comparisons are particularly unreliable. Since our task specifically involves designing a single binder for multiple distinct targets, reporting $\Delta G$ values across different binder–target pairs could be more misleading than informative.
>
> For these reasons, we adopted TM-score (novelty) as the metric for relating designed proteins to real-world data, and we believe it is more appropriate than RMSD or $\Delta G$ for our setting.
>
> ---
>
> **[W2 & Q2] Comparison with traditional diffusion methods.**
>
> We would like to clarify that I3CD addresses the **single-state binder co-design** problem, generating both the sequence and structure of a binder given a target's sequence and structure, which is distinct from unconditional single-protein co-design problems (e.g., Multiflow). Within the binder co-design setting, we have compared I3CD against APM, a state-of-the-art binder co-design method, in Appendix F of the manuscript.
>
> In addition, we have now included baselines for the cross-context binder design task, including one based on RFDiffusion (a diffusion-based backbone generation method). Please refer to our response to Reviewer EyZZ [W2] for full details and quantitative results.
>
> ---
>
> **[W3 & Q3] The scoring function.**
>
> The design of our scoring function was informed by two sources: the reward used in Complexa and the quality filtering criteria we employed during dataset construction. Regarding the hyperparameter choices, we adopted a heuristic weighting scheme with $\omega_{\text{ipae}} = 0.5$, $\omega_{\text{plddt}} = 0.3$, and $\omega_{\text{rmsd}} = 0.2$. This weighting deliberately emphasizes the inter-chain interaction quality (ipAE), thereby steering beam search toward designs with stronger binder–target interactions.
>
> We note that our framework is fully compatible with alternative scoring functions or different hyperparameter configurations. These would simply redirect the optimization direction of beam search and can be adjusted based on specific practical requirements. Since the primary contribution of this work is to propose a method for the entirely new task of cross-context binder design, we consider the exploration of optimal scoring functions an interesting direction for future work and will note this in the revised manuscript.
>
> ---
>
> **[Q4] Training computation cost comparison.**
>
> We provide a comparison of training costs between our method and APM below:
>
> | Method | Training Steps         | GPUs     |
> |:------:|:----------------------:|:--------:|
> | APM    | 257k (stage 1)         | 64× H100 |
> | APM    | 837k (stage 1)         | 8× H100  |
> | APM    | 235k (stage 2)         | 64× H100 |
> | Ours   | 715k (single stage)    | 8× A100  |
>
> Our training computation cost is approximately 15% of APM's, while our model's parameter count is roughly 11% of APM's (as detailed in Appendix F). Despite this significantly reduced computational budget, our method achieves comparable performance on single-state binder design (Table 3), demonstrating the efficiency of our approach.

---

> > ### Author Rebuttal · Reviewer_XNNu · 2026-04-03
> >
> > The justification for avoiding RMSD and Rosetta ΔΔG is largely argumentative rather than empirical, leaving the lack of real-world validation still unconvincing. The scoring function hyperparameters are admitted to be heuristic with no ablation, and the I3CD comparison against joint-noising diffusion baselines is deflected by pointing to a different comparison rather than directly addressed.

---

> > > ### Author Response · Authors · 2026-04-05
> > >
> > > We sincerely thank the reviewer for the continued engagement. We have conducted additional experiments to address every remaining concern below.
> > >
> > > ---
> > >
> > > **[W1 & Q1]: Additional Metrics (RMSD and $\Delta \Delta G$)**
> > >
> > > Although the TM-score-based novelty metric already provides a comparison against real-world data (Table 1), we have now computed both RMSD (against the closest PDB matches) and Rosetta $\Delta \Delta G$. Results are shown below:
> > >
> > > | ipAE_0 | binder_pLDDT_0 | binder_scRMSD_0 | Unique_Success_0 | novelty_0 | RMSD_0 | $\Delta \Delta G$_0 | ipAE_1 | binder_pLDDT_1 | binder_scRMSD_1 | Unique_Success_1 | novelty_1 | RMSD_1 | $\Delta \Delta G$_1 | Both_Success |
> > > | - | - | - | - | - | - | - | - | - | - | - | - | - | - | - |
> > > | 4.32 | 90.6 | 1.96 | 8 | 0.437 | 10.6 | -9.40$\pm$40.2 | 4.23 | 88.7 | 2.07 | 10 | 0.588 | 6.86 | -13.7$\pm$68.1 | 7 |
> > >
> > > **RMSD.** We use Foldseek to search the PDB database for proteins that match the designed binders and compute RMSD accordingly. If no matching protein is found, a default value of RMSD = 20 is assigned. The RMSD metric is consistent with the novelty (TM-score) metric: high RMSD corresponds to low novelty scores, indicating that the binders designed by our method exhibit good structural novelty.
> > >
> > > **$\Delta \Delta G$.** The computed mean values show good consistency with metrics such as pLDDT. However, due to the instability of energy-related steps such as relaxation, the standard deviations tend to be large. Moreover, prior work [1] has shown that for binder design tasks, $\Delta \Delta G$ is less informative than confidence-based metrics such as pLDDT.
> > >
> > > ---
> > >
> > > **[W3 & Q3]: Scoring Function Hyperparameter Ablation**
> > >
> > > We have conducted the requested ablation study. Results are shown below:
> > >
> > > | w_ipae | w_plddt | w_rmsd | ipAE_0 | binder_pLDDT_0 | binder_scRMSD_0 | Unique_Success_0 | novelty_0 | ipAE_1 | binder_pLDDT_1 | binder_scRMSD_1 | Unique_Success_1 | novelty_1 | Both_Success |
> > > | - | - | - | - | - | - | - | - | - | - | - | - | - | - |
> > > | 1.0 | 0.0 | 0.0 | 4.75 | 87.5 | 2.14 | 11 | 0.335 | 4.88 | 86.8 | 1.80 | 11 | 0.335 | 10 |
> > > | 0.0 | 1.0 | 0.0 | 4.36 | 90.2 | 2.05 | 8 | 0.597 | 4.77 | 90.2 | 1.73 | 9 | 0.424 | 7 |
> > > | 0.0 | 0.0 | 1.0 | 5.98 | 85.7 | 1.80 | 8 | 0.336 | 5.17 | 87.4 | 1.41 | 11 | 0.174 | 7 |
> > > | 0.33 | 0.33 | 0.33| 4.84 | 89.2 | 1.82 | 10 | 0.454 | 4.89 | 88.4 | 2.16 | 12| 0.385 | 10 |
> > > | 0.5 | 0.3 | 0.2 | 4.32 | 90.6 | 1.96 | 8 | 0.437 | 4.23 | 88.7 | 2.07 | 10 | 0.588 | 7 |
> > >
> > > Shifting weight toward a specific metric generally improves that metric, but may degrade others. While different weight configurations affect the final both success count, extreme settings such as (1.0, 0.0, 0.0) achieve a higher both success count yet exhibit noticeably weaker scRMSD and pLDDT compared to most other configurations. Our chosen setting balances high quality across all three metrics (ipAE, binder_pLDDT, binder_scRMSD) with a reasonable number of both success, and, as intended by the weight design, achieves strong ipAE performance. We will include this ablation in the revised manuscript.
> > >
> > > ---
> > >
> > > **[W2 & Q2]: Comparison with Joint-Noising Diffusion Baselines**
> > >
> > > We respectfully clarify three points:
> > >
> > > **1. I3CD is designed to serve cross-context binder design, not to be an independently optimal single-state binder design method.** The primary contribution of our work is the cross-context binder design paradigm. In I3CD, clean target sequences and structures are provided as conditioning signals to guide binder generation, while the denoising process for binder design decouples sequence and structure, facilitating subsequent sequence-conditioned binder denoising in MoPS. All design choices in I3CD serve this overarching goal. Evaluating I3CD solely as a standalone single-state method does not reflect its intended purpose.
> > >
> > > **2. To our knowledge, no open-source diffusion-based binder co-design model existed as of our submission deadline** that could serve as a direct "joint-noising diffusion baseline". If the reviewer could point to a specific method, we would be happy to include such a comparison in the revised revisions.
> > >
> > > **3. Despite the absence of diffusion-based binder co-design baselines,** we have provided a comparison between I3CD and APM (a state-of-the-art binder co-design method) in Appendix F.
> > >
> > > ---
> > >
> > > [1]  Bennett, N. R. et al. Improving de novo protein binder design with deep learning. Nat Commun 14, 2625 (2023).

---

### Official Review · Reviewer_1WGH · 2026-03-18

**Soundness:** 3
**Presentation:** 3
**Significance:** 3
**Originality:** 2
**Overall Recommendation:** 4
**Confidence:** 4

**Summary:**

This paper studies how to design a single protein binder that can work across multiple contexts, especially different conformations. The authors propose Chamaileon, which keeps the sequence consistent while allowing the structure to adapt, and uses an alternating optimization strategy (MoPS) together with beam search to balance multiple binding requirements. They evaluate the method on a newly constructed benchmark and show that it can produce binders that remain effective across contexts.

**Compliance With Llm Reviewing Policy:**

Affirmed.

**Final Justification:**

The authors’ additional information has addressed most of my concerns, so I have adjusted my score accordingly.

**Key Questions For Authors:**

1. It would be worth clarifying whether a stronger baseline could be established based on existing methods. For instance, by using RFdiffusion in conjunction with ProteinMPNN to generate a set of candidates, and then applying strategies such as beam search or specific filtering criteria to select designs that demonstrate robust performance across multiple target proteins. While such a pipeline is technically feasible, no such attempts were reported in the present study; could the authors comment on this?

2. Could the authors include a comparative analysis against existing methods within a single-target setting? This would provide a clearer perspective on the performance of the base model trained in this manner.

3. Could the authors provide results evaluated under more stringent criteria (e.g., scRMSD < 2.0 A)? This would help facilitate a more objective assessment of the method's performance in practical binder design scenarios.

**Limitations:**

Yes

**Strengths And Weaknesses:**

Strengths

1. This work introduces a new problem formulation that extends traditional single-structure protein design to settings involving multiple conformations and even multiple targets. This direction is well motivated, as it better reflects the dynamic nature of proteins and the need for multi-specific binding in real biological systems.

2. On the methodological side, the use of MoPS and beam search provides a practical way to balance multiple constraints at inference time, without requiring additional multi-state training data. The overall approach is relatively simple and easy to implement, while still leading to clear improvements in the reported results.

Weaknesses

1. The paper appears to make somewhat exaggerated claims regarding its multi-target capabilities. Although "multi-target" is frequently mentioned in the problem formulation and analysis, there is no dedicated quantitative evaluation for this setting; the primary benchmark (CROSS) is constructed from multi-state systems. Given that multi-target scenarios often involve larger structural and interface divergence, relying mainly on qualitative examples (e.g., Figure 5) makes it difficult to assess the method's effectiveness in this setting.

2. Furthermore, the experiments do not include comparisons against existing state-of-the-art methods (such as RFdiffusion or ProteinMPNN), and instead focus primarily on internal ablations. While the authors note that existing methods are not directly designed for cross-context settings, stronger adapted baselines would still help clarify the relative performance gains.

3. There are also questions regarding the evaluation criteria. The paper adopts an scRMSD threshold of < 5.0 A  as part of its success definition, whereas stricter thresholds (e.g., < 2.0 A) are often used in binder design to indicate high structural accuracy. The use of a more lenient threshold may overestimate success rates, and it would be helpful to report results under stricter criteria.

---

> ### Author Rebuttal · Authors · 2026-03-31
>
> We thank the reviewer for the thoughtful feedback and for recognizing the value of our new problem formulation and the simplicity and effectiveness of our method. We address each concern below.
>
> ---
>
> **[W1] Multi-target scenarios.**
>
> We apologize for any misunderstanding. CROSS was not intended to be exclusively a multi-state benchmark; we explicitly increased the proportion of entries with higher target divergence during construction (Lines 366–373).
>
> To be precise, **multi-target** refers to designing a binder that binds two structurally and sequentially distinct targets, while **multi-state** refers to binding different conformations of the same (or nearly identical) target. The key distinction lies in sequence identity, which manifests as varying degrees of structural divergence. Using a 95% target sequence identity threshold to separate the two, the composition of CROSS is as follows:
>
> |              | Number |
> |:-------------|:------:|
> | Multi-target | 15     |
> | Multi-state  | 85     |
>
> CROSS explicitly covers both settings, and among our successful cases, one is indeed a multi-target success:
>
> | Target Identity | ipAE_0 | pLDDT_0 | scRMSD_0 | ipAE_1 | pLDDT_1 | scRMSD_1 |
> |:---------------:|:------:|:-------:|:--------:|:------:|:-------:|:--------:|
> | 94%             | 3.32   | 93.6    | 0.90     | 3.16   | 94.1    | 1.17     |
>
> This accounts for 1 of 7 both-success cases (14.3%), close to the multi-target proportion in CROSS (15%). Among all 1,867 candidates before filtering:
>
> |              | Number |
> |:-------------|:------:|
> | Multi-target | 200    |
> | Multi-state  | 1,667  |
>
> The multi-target proportion among all candidates (10.7%) is lower than in CROSS (15%), indicating that (1) multi-target data is naturally scarce and that (2) we deliberately enriched CROSS to better cover multi-target scenarios.
>
> We emphasize that multi-target binder design is an inherently challenging task with no existing end-to-end approach, and even naive adaptations exhibit fundamental limitations (as shown by the two baselines introduced in our response to [W2&Q1]). Our work provides a feasible solution with a successful multi-target case as a proof of concept. However, we acknowledge considerable room for improvement, given the scarcity of evaluation data, our relatively small model size, and the potentially limited capability of AF2 itself in assessing multi-target binding. We are actively preparing more comprehensive multi-target benchmarks for future versions.
>
> ---
>
> **[W2 & Q1] Lack of baselines.**
>
> We appreciate this suggestion and have now included two baselines. Please refer to our response to Reviewer EyZZ [W2] for the full details.
>
> ---
>
> **[W3 & Q3] Stricter scRMSD threshold.**
>
> We provide results under the stricter scRMSD < 2.0 Å criterion below:
>
> |          | ipAE_0 | pLDDT_0 | scRMSD_0 | success_0 | ipAE_1 | pLDDT_1 | scRMSD_1 | success_1 | both_success |
> |:---------|:------:|:-------:|:--------:|:---------:|:------:|:-------:|:--------:|:---------:|:------------:|
> | scRMSD < 5 Å | 4.32 | 90.6 | 1.96 | 8 | 4.23 | 88.7 | 2.07 | 10 | 7 |
> | scRMSD < 2 Å | 4.47 | 91.4 | 1.20 | 5 | 3.82 | 91.8 | 1.48 | 7  | 4 |
>
> Even under this stricter threshold, Chamaileon retains a meaningful number of successful designs, and the average scRMSD values of the passing designs are notably low (1.20 Å and 1.48 Å), indicating high structural accuracy.
>
> Furthermore, we would like to highlight that MoPS combined with beam search naturally admits a **computation vs. quality trade-off**: by increasing the computational budget, one can obtain higher-quality designs that pass more stringent thresholds. Notably, under our default setting (beam search candidate number = 4, frequency = 50), sampling a single design on one NVIDIA A100 GPU takes only approximately **3.97 minutes**, whereas Complexa can require up to 4 hours per design (as reported in Appendix F of Complexa). To illustrate this trade-off, we performed 10 design runs on a single entry with varying beam search hyperparameters. The mean metrics are shown below:
>
> | Candidate Num | Freq | ipAE_0 | pLDDT_0 | scRMSD_0 | ipAE_1 | pLDDT_1 | scRMSD_1 |
> |:-------------:|:----:|:------:|:-------:|:--------:|:------:|:-------:|:--------:|
> | 1             | 500  | 13.5   | 57.4    | 5.56     | 13.7   | 56.3    | 3.28     |
> | 2             | 250  | 16.2   | 49.1    | 4.71     | 14.8   | 51.4    | 4.10     |
> | 4             | 125  | 14.9   | 55.1    | 3.93     | 15.1   | 53.2    | 4.91     |
> | 8             | 50   | 10.0   | 71.1    | 3.92     | 6.21   | 79.4    | 3.69     |
>
> As the computational budget increases (with correspondingly finer-grained beam search), design quality improves substantially across all metrics, confirming that Chamaileon can achieve stronger results with additional computation while remaining far more efficient than existing alternatives.
>
> ---
>
> **[Q2] Single-target results.**
>
> We provide single-state binder design results in Appendix F of the manuscript.

---

> > ### Author Rebuttal · Reviewer_1WGH · 2026-04-03
> >
> > The authors’ additional information has addressed most of my concerns, so I have adjusted my score accordingly.

---

> > > ### Author Response · Authors · 2026-04-05
> > >
> > > We sincerely thank the reviewer for the constructive feedback throughout the review process. The suggestions and concerns raised have been invaluable in strengthening our work, and we are grateful for the time and effort dedicated to evaluating our manuscript. We will incorporate all discussed improvements into the revised version.

---

### Official Review · Reviewer_JJFr · 2026-03-25

**Soundness:** 3
**Presentation:** 3
**Significance:** 3
**Originality:** 2
**Overall Recommendation:** 5
**Confidence:** 4

**Summary:**

The authors propose Chamaileon, a framework for cross-context binder design, meaning that one can design both for multiple targets at the same time as well sa for mulitple sites/conformations of the same target. They leverage inference-time compute via their proposed Mixture-of-Paths samping procedure on top of the co-deisign model and then evaluate their model on the newly curate CROSS benchmark.

**Compliance With Llm Reviewing Policy:**

Affirmed.

**Final Justification:**

I increased my score since the authors addressed some of my concerns.

**Key Questions For Authors:**

[Q1]: the paper mentions off target binding (not C abut A and b), but this does not seem to be demonstrated anywhere. Why not, and how would you evualate it?

[Q2]: biophysical priors are mentioned in the conclusion, but what kind of priors you think would be interesting, and how should they be used in this setting?

**Limitations:**

yes

**Strengths And Weaknesses:**

[S1] clever inference time setup: the authors recognise the limitations that the multistate datasets have and therefore focus on generating multispecific binders at inference time.

[S2] Approach is scalable to a high number of potential conformations, which could potentially open up the way foo more complex generation scenarios.

[S3] Bioinformatic analysis: beyond just the mere benchmark metrics, the authors also try to interpret qualitatively different ways their model solves the problems.

[W1] The benchmark is quite small which makes it hard to draw robust conclusions. Can the authors extend the bechmark to either more data or ideally also extend it for example to small molecule bnder cases? Also it does not seem as if the benchmark is available as part of the submission, making it very hard to judge the soundness of the enw benchmark

[W2] The model does not compare to any baselines; while there are no explicitly trained models for this task, it has been attacked before with RFDiffusion and other models, so using some of the previously pubished approaches for these things as a benchmark would be useful.

---

> ### Author Rebuttal · Authors · 2026-03-31
>
> We sincerely thank the reviewer for the constructive feedback and for recognizing the strengths of our work, including the inference-time strategy that circumvents the scarcity of multi-state datasets (ours aiming for binders specifically), the scalability of MoPS, and the value of our bioinformatic analysis. We address each concern in detail below.
>
> ---
>
> **[W1] Benchmark size and availability.**
>
> We appreciate this concern. We would like to clarify that CROSS is curated through a rigorous pipeline: starting from 1,867 candidates, we applied target similarity-based category balancing and quality-based filtering to arrive at a compact yet high-quality benchmark. This design philosophy serves two purposes: (1) it substantially reduces the computational cost of evaluation, and (2) it ensures that each benchmark entry is of sufficient quality to yield reliable assessment.
>
> To further demonstrate the robustness of our framework, we conducted an additional evaluation on a larger subset of 1,000 entries sampled from the full 1,867 candidates. The results are shown below:
>
> | | ipAE_0 | pLDDT_0 | scRMSD_0 | success_0 | novelty_0 | ipAE_1 | pLDDT_1 | scRMSD_1 | success_1 | novelty_1 | both_success |
> |---|:---:|:---:|:---:|:---:|:---:|:---:|:---:|:---:|:---:|:---:|:---:|
> | Ours | 6.03 | 84.3 | 2.26 | 17 | 0.851 | 5.58 | 85.3 | 2.23 | 17 | 0.881 | 13 |
>
> The lower success rate compared to Table 1 in the main paper confirms that cross-context binder design becomes more challenging on lower-quality targets, which further validates the rationale behind our quality-based filtering for CROSS. We will include results on all 1,867 candidates in the camera-ready version.
>
> Due to policy restrictions, we are unable to release the raw data of CROSS during the rebuttal phase. We will open-source all related data upon acceptance of the paper.
>
> ---
>
> **[W2] Lack of baselines.**
>
> We appreciate this suggestion and have now included two strong baselines. Please refer to our response to Reviewer EyZZ [W2] for the full details, including the baseline construction methodology and quantitative comparison.
>
> ---
>
> **[Q1] Off-target binding evaluation.**
>
> To evaluate off-target binding, we selected four cases where Chamaileon successfully designed binders for both contexts (both_success) and can be retrieved an additional third conformation from CoDNAS, which was not included during the design process. We then assessed whether the binders designed for the original two conformations also bind to this unseen third conformation. Additionally, we tested Chamaileon with all three conformations provided as input during MoPS sampling. The averaged results are:
>
> | Setting | ipAE_0 | pLDDT_0 | scRMSD_0 | ipAE_1 | pLDDT_1 | scRMSD_1 | ipAE_2 | pLDDT_2 | scRMSD_2 |
> |---|:---:|:---:|:---:|:---:|:---:|:---:|:---:|:---:|:---:|
> | 2-conf. design | 3.54 | 92.2 | 1.09 | 4.56 | 89.2 | 1.49 | 22.2 | 74.1 | 5.31 |
> | 3-conf. design | 4.92 | 88.7 | 2.55 | 4.67 | 90.6 | 2.01 | 4.11 | 90.9 | 1.67 |
>
> The results clearly show that when the third conformation is not considered during sampling, the designed binders exhibit off-target binding behavior (high ipAE and low pLDDT for the third context). In contrast, when all three conformations are jointly considered via MoPS, the binders successfully engage all three targets. This demonstrates both the specificity of MoPS-designed binders and the controllability of our framework, users can precisely specify which contexts to target, and MoPS will optimize accordingly.
>
> ---
>
> **[Q2] Biophysical priors.**
>
> We appreciate this insightful question.
>
> **Regarding "what kind of priors"**: We primarily refer to the degree of regional structural fluctuation derived from conformational ensemble predictions or prior structural knowledge. A classic example is Class A GPCRs, where the TM6 helix undergoes an outward movement in the active state to accommodate downstream effectors like G proteins. In physiological environments, these active and inactive states naturally coexist in a dynamic conformational equilibrium.
>
> **Regarding "how they should be used"**: These priors can be integrated in further framework to define the entire conformational ensemble as the target condition, thereby providing more precise spatial constraints for interface complementarity across different states. For example, such approach would benefit a recent advancement, *De Novo Design of Exoframe Modulators*. While current GPCR exoframe modulators (GEMs) are successfully designed against a single static conformation, incorporating these biophysical priors to target the dynamic ensemble could enable the design of multi-state binders and achieve more precise conformational ensemble manipulation.
>
> [1] Cheng et. al., De novo design of GPCR exoframe modulators. Nature(2026)

---

> > ### Author Rebuttal · Reviewer_JJFr · 2026-04-03
> >
> > Thanks for the replies to my concerns. A few questions that still remain:
> > [Q1] My question was more related to doing active negative design. In your setting you simply do not include the third conformation and in this case this causes it to have bad metrics, but there might be settings where the two conformations are quite similar and negative design by just excluding the unwanted target might not work. Do you have a mechanism to explicitly avoid a specific conformation?
> >
> > [W2] Thanks for adding these baselines. The RFDiffusion + tied MPNN pipeline has been used in previous publications for wet-lab validated multi-state design, however here it does not give any successes in your benchmark. Any idea why this might be?
> >
> > [W1] You mention two things here in the paper: multi-state design and multi-target design. as far as I understand your benchmark here only tests one of these correct?

---

> > > ### Author Response · Authors · 2026-04-03
> > >
> > > We sincerely thank the reviewer for the continued engagement with our work. We address each remaining question below.
> > >
> > > ---
> > >
> > > **[Q1] Active negative design.**
> > >
> > > We apologize for the earlier misunderstanding. Our framework does support active negative design through a straightforward mechanism: we modify the score function used to rank candidates during beam search by negating the scores computed on the unwanted target, then aggregate all scores normally for ranking. Meanwhile, during MoPS, we only alternate between the two desired targets. The results are as follows:
> > >
> > > | Setting | ipAE_0 | pLDDT_0 | scRMSD_0 | ipAE_1 | pLDDT_1 | scRMSD_1 | ipAE_2 | pLDDT_2 | scRMSD_2 |
> > > |---|:---:|:---:|:---:|:---:|:---:|:---:|:---:|:---:|:---:|
> > > | 3-conf. design | 4.92 | 88.7 | 2.55 | 4.67 | 90.6 | 2.01 | 4.11 | 90.9 | 1.67 |
> > > | 2-conf. design | 3.54 | 92.2 | 1.09 | 4.56 | 89.2 | 1.49 | 22.2 | 74.1 | 5.31 |
> > > | (2-act. & 1-neg.)-conf. design | 5.10 | 88.2 | 1.81 | 4.51 | 90.5 | 3.10 | 24.0 | 73.5 | 5.53 |
> > >
> > > With the active/negative sampling strategy, the designed binder exhibits weaker binding to the third (unwanted) target: ipAE, pLDDT, and scRMSD on target 2 all degrade compared to the 2-conformation design baseline, confirming that this mechanism effectively repels the unwanted conformation. This demonstrates that Chamaileon can be readily extended to support explicit negative design without architectural modifications.
> > >
> > > ---
> > >
> > > **[W2] Failure analysis of Baseline 1 (RFDiffusion + ProteinMPNN).**
> > >
> > > While prior work has demonstrated wet-lab validated multi-state design of **single-chain** proteins using similar RFDiffusion + ProteinMPNN pipelines [1, 2], we identify three key factors that likely explain the failure of our Baseline 1. First, [1] and [2] focus on single-chain multi-conformation design, whereas cross-context binder design is a substantially harder task that requires designing a protein under distinct target constraints, significantly increasing the difficulty. Second, [1] sampled 10,000+ sequences via ProteinMPNN while [2] sampled 200+ sequences resulting 1000+ conformations (under a simplified setting) for an individual design, whereas our Baseline 1 sampled 4 sequences for each entry to align with Chamaileon's beam search candidate number of 4 for a fair comparison; the drastically smaller sample size naturally reduces the likelihood of finding successful designs. Third, these methods introduce additional operational steps, such as Rosetta-based design filtering, rather than following a straightforward pipeline like RFDiffusion + ProteinMPNN.
> > >
> > > ---
> > >
> > > **[W1] Multi-state and multi-target design in CROSS.**
> > >
> > > CROSS covers both multi-state and multi-target scenarios. Given the inherent imbalance in available data, we manually adjusted the proportions to ensure adequate coverage of both settings. We refer the reviewer to our response to Reviewer 1WGH [W1] for a detailed breakdown of the benchmark composition.
> > >
> > > ---
> > >
> > > [1] Guo, A. B., Akpinaroglu, D., Kelly, M. J. S. & Kortemme, T. Deep learning guided design of dynamic proteins. Preprint at https://doi.org/10.1101/2024.07.17.603962 (2024).
> > >
> > > [2] Praetorius, F. et al. Design of stimulus-responsive two-state hinge proteins. (2023).

---

### Decision · Program_Chairs · 2026-04-30

**Decision:**

Accept (spotlight)

**Comment:**

While most generative protein binder design methods produce "static" proteins binding a single target in a single conformation, Chamaileon tackles challenging multi-state and multi-target design tasks, a valuable and impactful departure from most prior works. The approach is smartly developed for that task, leveraging appropriate inference-time strategies to guide generation towards desired multi-state binders, binding different targets or epitopes. For empirical validation, the authors introduce a new dataset and benchmark, CROSS, which will be valuable to the community. Extensive experiments support the proposed approach.

All reviewers suggest acceptance. The authors were able to address the reviewer's questions and they provided additional experiments and analyses during the rebuttal. I think the paper's contributions are significant and I agree with the reviewers' positive assessments. Therefore, I recommend the paper for acceptance.

Importantly, I would like to ask the authors to release the CROSS benchmark and dataset for the community, as was promised during the rebuttal phase.